# Nanofiber-interwoven gel membranes with tunable 3D-interconnected transport channels for efficient CO$_2$ separation

Hao-Nan Li [1,2,4], Ze-Yu Sun[1,2,4], Zhen-Jie Yu[1,2], Kexin Man[1,2], Chao Zhang [1,2] & Zhi-Kang Xu [1,2,3]

Mixed matrix membranes (MMMs) capable of breaking the permeability-selectivity trade-off suffer from the inefficient and disconnected bulky transport channels as well as inferior interfacial compatibility between nanomaterials and polymers. Herein, we propose an original photothermal-triggered in-situ gelation approach to elaborate an original class of MMMs, termed nanofiber-interwoven gel membranes (NIGMs) that feature tunable 3D-interconnected ultrafast transport channels and highly-selective CO$_2$-philic gel for boosting CO$_2$ separation performance. The key design of NIGMs lies in leveraging dual functions of CNT-interwoven skeleton: (1) serving as a photothermal confined reactor that rapidly triggers in-situ gelation of highly-selective CO$_2$-philic gel without phase separation-induced interfacial defects to construct defect-free and thickness-controllable NIGMs; (2) functioning as a 3D-interconnected continuous skeleton for providing ultrafast CO$_2$ transport channels. By orchestrating the distribution and configuration of interwoven nanofibers, the NIGMs possess a boosted CO$_2$ permeance of 211.0 GPU increased by 1558% over polymeric gel counterparts and an ultrahigh CO$_2$/N$_2$ and CO$_2$/CH$_4$ selectivity of up to 151 and 47 respectively. Our work offers a paradigm shift in developing advanced MMMs beyond gas separation.

Mixed matrix membranes (MMMs), characterized by collective merits of polymer-enabled excellent processability and nanomaterial-enabled superior selective transport channels, have emerged as energy-efficient and environmentally-sustainable candidates to improve separation performance that goes beyond conventional polymer membranes[1–3]. These MMMs hold great potential in a myriad of separation scenarios spanning from ion screening, molecule sieving, and gas separation to energy harvesting[4,5]. Over the past decade, tremendous efforts have been made in exploiting various advanced nanomaterials—impermeable and permeable ones from 0D nanoparticles to 1D nanofibers and 2D nanosheets, as functional additives in

MMMs for attaining preferential separation performance[6–8]. The key to successfully translating distinct properties of these nanomaterials into MMMs lies in manipulating the spatial distribution of nanomaterials within polymer matrix to form fine transport channels[9,10]. However, most MMMs are perplexed with the easy formation of unwanted non-selective transport channels owing to inevitable agglomeration and sedimentation of nanomaterials in polymer matrix as well as poor compatibility between nanomaterials-polymer interfaces[11,12]. Furthermore, these nanomaterials localized in bulky polymer matrix tend to be discrete and disconnected, making bulky transport channels inconsecutive and inefficient[13,14]. One of the most effective approaches

[1]MOE Key Laboratory of Macromolecular Synthesis and Functionalization, and Key Laboratory of Adsorption and Separation Materials & Technologies of Zhejiang Province, Department of Polymer Science and Engineering, Zhejiang University, Hangzhou, China. [2]The "Belt and Road" Sino-Portugal Joint Lab on Advanced Materials, International Research Center for X Polymers, Zhejiang University, Hangzhou, China. [3]Institute of Marine Chemistry and Environment, Ocean College, Zhejiang University, Zhoushan, China. [4]These authors contributed equally: Hao-Nan Li, Ze-Yu Sun. ✉e-mail: zhangchao7@zju.edu.cn; xuzk@zju.edu.cn

to construct continuous transport channels is to boost the loading content of nanomaterials within the polymer matrix[15,16]. Nevertheless, this strategy is only suitable for a few material systems with matching physicochemical properties of nanomaterials and polymers[6,17]. Moreover, under high loading content, the majority of nanomaterials would encounter more pronounced agglomeration and sedimentation for generating non-selective voids and pinholes[18,19]. Therefore, it remains long-standing challenge to overcome these limitations through conventional strategies and it is highly essential to develop approach to construct MMMs with interconnected transport channels.

We propose a design strategy to fabricate a class of MMMs with ultrafast transport channels. As a core component of the hearts, the 3D densely-stacked cardiac muscle fibers have evolved as an indispensable platform to rapidly propagate signals in a high-fidelity manner along the direction of cardiac muscle fibers[20,21]. Inspired by this, we present an original architecture of nanofiber-interwoven gel

membranes (NIGMs) consisting of CNT-interwoven tunable 3D-interconnected $CO_2$ transport highways and highly-selective $CO_2$-philic gel matrix, enabling a prominent boost in both permeance and selectivity towards $CO_2$ separation (Fig. 1a). The NIGMs are formulated by an innovative photothermal-triggered in-situ gelation method, in which the CNT-interwoven skeleton is harnessed as a photothermal confined reactor to create highly-crosslinked $CO_2$-philic gel exhibiting an interpenetrated architecture with CNT-interwoven skeleton. The in-situ formed $CO_2$-philic gel matrix has decent interfacial compatibility with CNT-interwoven skeleton without the occurrence of non-selective defects, meanwhile harnessing its abundant polar ethylene oxide units to facilitate $CO_2$ solubility over $N_2$ for attaining enhanced selectivity. Moreover, the distribution, stacking and configuration of interwoven nanofibers in our NIGMs can be easily regulated for creating 3D-interconnected continuous skeleton as ultrafast transport pathways to boost $CO_2$ permeance. We demonstrate that the NIGMs deliver an

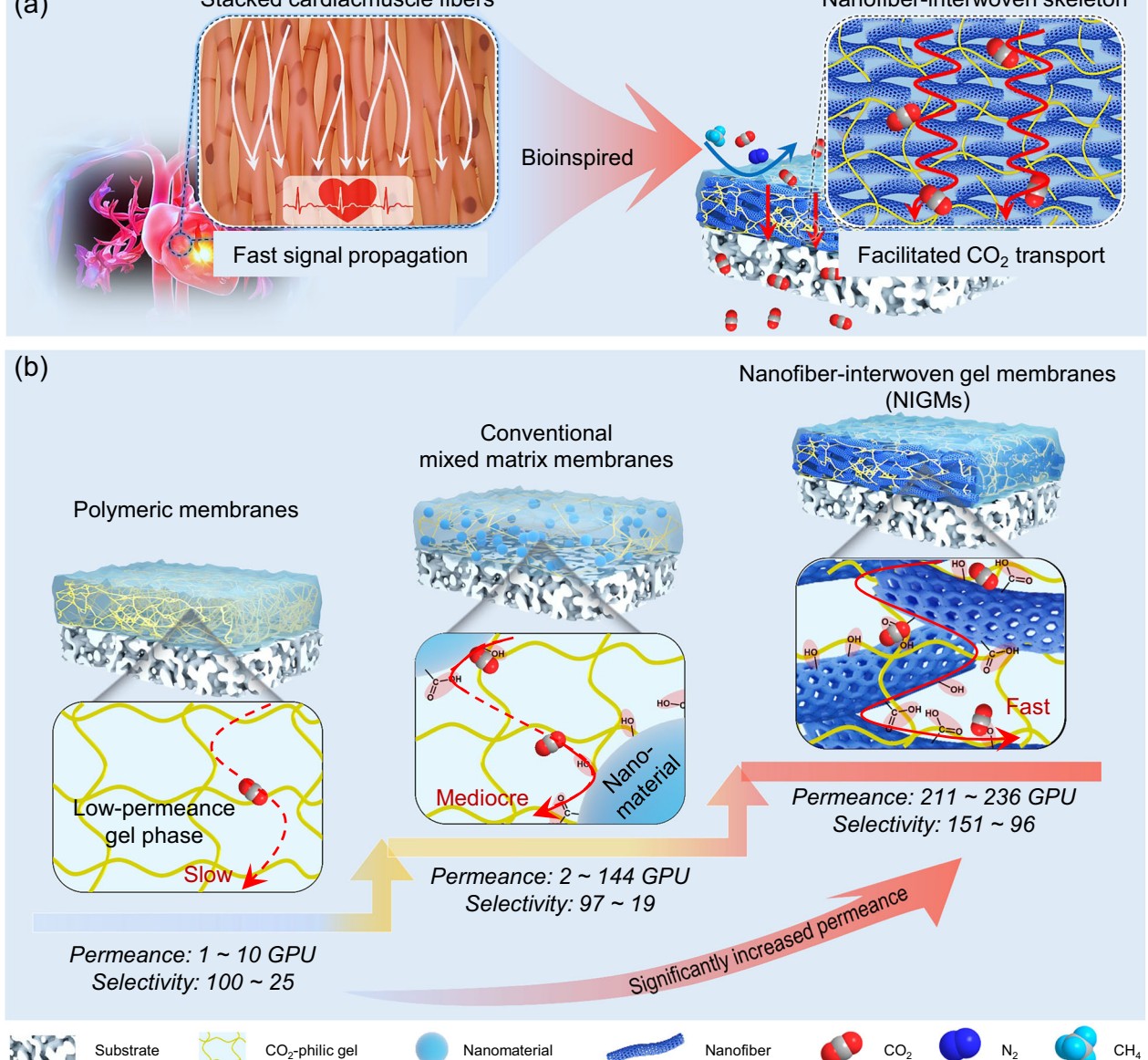

**Fig. 1 | Design concept and merits of nanofiber-interwoven gel membranes (NIGMs). a** Bio-inspired design of NIGMs for efficient $CO_2$ separation. Learning from rapid propagation of signals along the 3D densely-stacked cardiac muscle fibers, NIGMs are imparted with CNT-interwoven interconnected transport channels and highly-selective $CO_2$-philic gel matrix. **b** Schematic illustration of structure and performance evolution of NIGMs with conventional polymeric membranes, including polyimide[22,23], cellulose acetate[24,25], polysulfone[25,26] and polycarbonates[25,27]. As comparisons, NIGMs show a prominent boost in permeance and selectivity simultaneously, both of which are much better than that of conventional counterparts.

excellent $CO_2$ permeance of 211.0 GPU and an ultrahigh $CO_2/N_2$ and $CO_2/CH_4$ selectivity of up to 151 and 47 respectively, outperforming most of conventional counterparts. Furthermore, our NIGMs can leverage the anchoring utility of rigid CNT-interwoven skeleton on polymeric gel matrix to realize impressive long-term service stability with a slight attenuation in both permeance and selectivity even after 480 h operation. The present study offers additional flexibility and dimension in exploiting nanofiber-interwoven membranes towards advanced separations.

## Results

### Design concept of NIGMs

The NIGMs were designed by extracting inspiration from the unique architecture of the cardiac muscle that features the 3D densely-stacked cell fibers for rapid signal propagation (Fig. 1a). From the perspective of architecture, the NIGMs are made of CNT-interwoven rigid skeleton and highly-selective $CO_2$-philic gel matrix, in which they have intact and defect-free interfaces even under high loading nanomaterials via topological polymer chain entanglement and multiple non-covalent interactions. One of the biggest architecture differences between our NIGMs and existing MMMs lies in the spatial distribution of nanomaterials within the polymer matrix. Unlike the discrete and discontinuous nanomaterials in conventional MMMs, the distribution, stacking and configuration of interwoven nanofibers in our NIGMs can be easily regulated for creating 3D-interconnected continuous skeleton as ultrafast transport pathways to boost $CO_2$ permeance. Moreover, the polymeric skeleton of our NIGMs is also different from traditional polymeric membranes that suffer from neither dense polymer chain packing nor limited $CO_2$ transport carriers. The NIGMs not only are made of flexible crosslinked gel networks with good chain mobility for enlarged fractional free volume, but also consist of numerous functional carriers to accelerate $CO_2$ transport, both of which lead to preferential $CO_2$ solution and diffusion over $N_2$ for attaining enhanced selectivity. Consequently, our NIGMs are imparted with preferential selectivity and permeance in $CO_2$ separation at the same time, and particularly feature even dozens of times higher $CO_2$ permeance than that of traditional polymeric membranes, including polyimide[22,23], cellulose acetate[24,25], polysulfone[25,26] and polycarbonates[25,27] (Fig. 1b). Most importantly, the NIGMs are capable of harnessing rigid and high-loading CNT-interwoven skeleton to form topological chain entanglement and multiple interactions with polymeric gel matrix, thereby anchoring polymeric gel matrix for the attainment of excellent structure and performance stability.

### Fabrication and structure characterization of NIGMs

Figure 2a shows that the NIGMs were fabricated using a facile photothermal-triggered in-situ gelation strategy and were implemented in a photothermal confined reactor made of densely-stacked CNT-based nanofibers. Specifically, the photothermal confined reactor was first constructed by steering the assembly and stacking of CNT into a nanofiber-interwoven skeleton onto the surface of porous substrate through vacuum filtration. Subsequently, the as-prepared CNT-interwoven photothermal confined reactor was infused with gel precursor comprising polyethylene glycol (PEG) as both monomer and cross-linker and ammonium persulfate as thermal initiators. Notably, this confined reactor can be used to precisely regulate the thickness of the infused gel precursor and prevent the infused gel precursors from entering the underneath porous membrane. Here, due to strong polar attraction between its ethylene oxide units and $CO_2$, PEG was selected as $CO_2$-philic polymeric gel matrix. Upon solar irradiation, the CNT-interwoven photothermal confined reactor leverages classic photothermal effect of CNT to realize localized heating and trigger thermal initiators for generating numerous free radicals, allowing gel precursor for in-situ formation of defect-free PEG-based $CO_2$-philic gel matrix. This photothermal-triggered in-situ gelation strategy doesn't involve

phase separation and thereby is capable of eliminating interfacial defects between nanomaterials and polymers. In contrast, the construction of conventional MMMs undergoes solvent evaporation accompanied with phase separation, posing a great risk in forming some non-selective defects at the interface between nanomaterials and polymers[17,20].

To manifest the feasibility of photothermal-triggered gelation, we first investigated the temperature variation of the CNT-interwoven photothermal confined reactor as a function of sunlight irradiation time. As depicted in Fig. 2b, the CNT-interwoven photothermal confined reactor exhibits a rapid response to sunlight and has a significant temperature increase (Supplementary Fig. 1). Upon 2 min sunlight irradiation, the surface temperature of CNT-interwoven photothermal confined reactor rapidly reaches up to 65.4 °C, implying remarkable heating efficiency originating from excellent photothermal effect of CNT-interwoven skeleton. Such a high photothermal-triggered temperature lays the foundation for replacing the conventional heating to induce thermal initiators for generating numerous free radicals, thereby allowing for initiating the polymerization and gelation of PEG monomers as evidenced by the disappearance of unsaturated double bonds of PEG monomers (Supplementary Fig. 2). The utility of this CNT-interwoven photothermal-triggered gelation can be reflected by examining the conversion of PEG monomers. Figure 2c indicates that the conversion ratio of monomers in the CNT-interwoven reactor can reach up to 60.4% within 3 min. This conversion ratio can be further improved by adjusting sunlight intensity and irradiation time. This photothermal-triggered high conversion ratio enables a high-efficient gelation for the formation of highly-crosslinked gel networks interpenetrated with CNT-interwoven skeleton.

The distinctive merit of photothermal-triggered in-situ gelation is to perfectly eliminate interfacial defects between nanomaterials and polymers even under high loading nanomaterials. As illustrated in Fig. 2d, the CNT-interwoven skeleton possesses a high stacking density to generate 3D-interconnected pores. Following the photothermal-triggered in-situ gelation process, the CNT-interwoven skeleton within NIGMs is uniformly coated with a dense polymeric gel matrix layer, without observable voids or defects (Fig. 2e). The nitrogen adsorption-desorption isotherms results reveal that NIGMs exhibit typical Type II isotherms with a BET surface area value of merely $1.6251 \, m^2 \, g^{-1}$, confirming that the NIGMs possess a relatively dense and defect-free structure (Supplementary Fig. 3). Owing to the high-stacking density of CNT-interwoven skeleton, the dense polymeric gel layer can be conformal with CNT-interwoven skeleton with a thickness of 4 μm without penetrating into underneath porous membrane (Fig. 2f). Moreover, this photothermal-triggered in-situ gelation method can be easily scaled up for constructing 30 cm-diameter NIGMs with uniform thickness of about 4 μm (Supplementary Figs. 4–7), implying the potential for large-scale fabrication of NIGMs.

Furthering inspecting the relative content of CNT-interwoven skeleton and gel layer, we found that the resulting defect-free NIGMs show an ultrahigh nanomaterial loading containing a CNT content of up to 66.7% (Supplementary Fig. 8), which is much higher than that of conventional MMMs exhibiting a nanomaterial loading of lower than 40%. This can be attributed to the fact that our photothermal-triggered in-situ gelation strategy circumvents phase separation that usually occurs in the fabrication of conventional MMMs, thereby eliminating interfacial defects and non-selective voids. To reveal the distribution of the CNT-interwoven skeleton within the NIGMs, transmission electron microscopy with zone slicing was employed. The CNT-interwoven skeleton of NIGMs is completely encapsulated by polymeric gel matrix with an interpenetrated architecture (Fig. 2f). These results all imply that photothermal-triggered in-situ gelation approach can be successfully implemented to construct defect-free NIGMs that consist of CNT-interwoven interconnected skeleton and polymeric gel matrix.

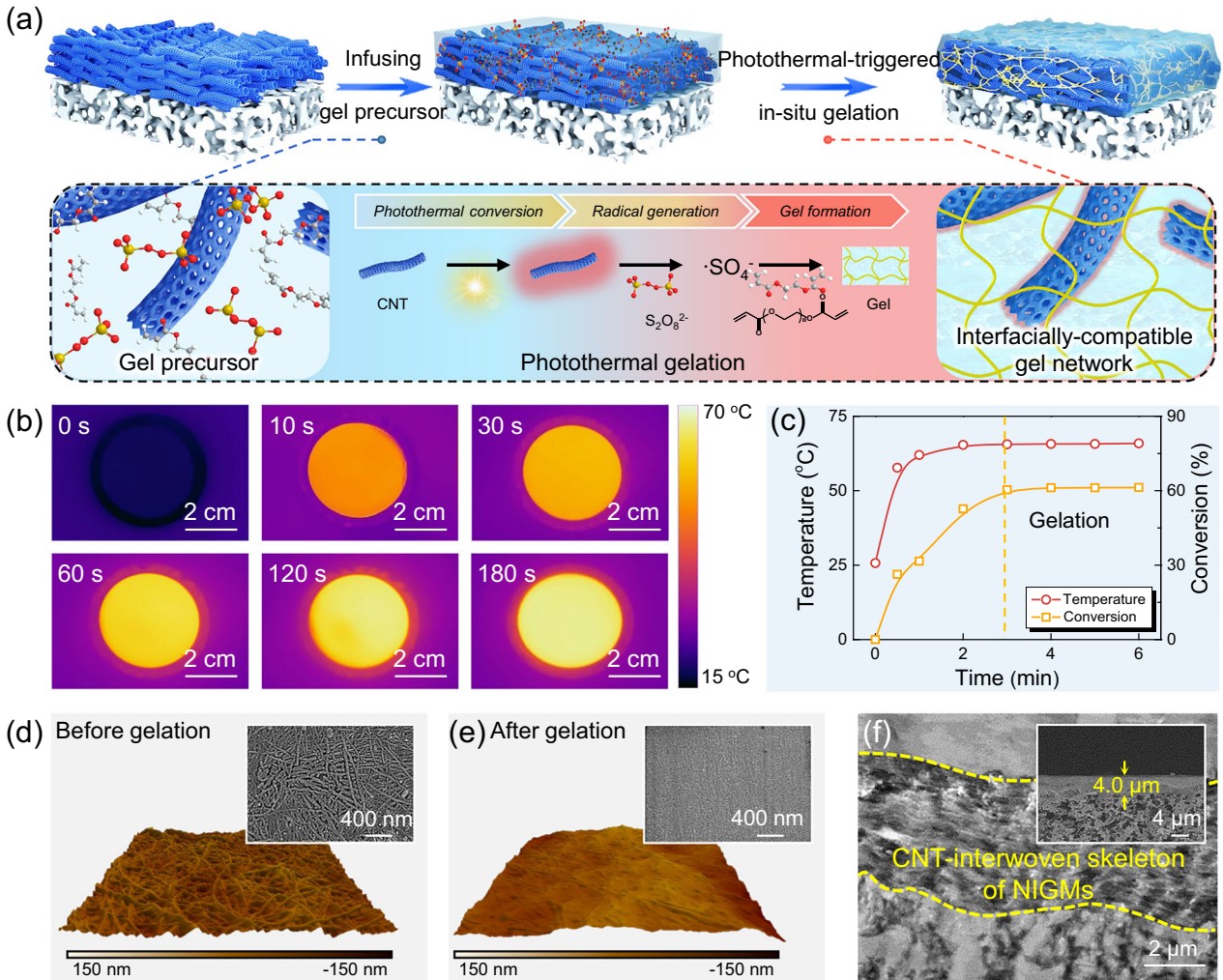

**Fig. 2 | Fabrication and structure characterization of NIGMs. a** Schematic illustration of the fabrication process of NIGMs by a facile photothermal-triggered in situ gelation method. **b** Infrared images of the CNT-interwoven photothermal confined reactor as a function of sunlight irradiation time. **c** Dynamic curve of temperature and monomer conversion ratio over reaction time for the fabrication of NIGMs via photothermal-triggered in situ gelation method. **d**, **e** AFM images and corresponding SEM images of surface morphologies of the densely-packed CNT-interwoven skeleton and the associated NIGMs. **f** Cross-sectional TEM and SEM images of the NIGMs. The CNT-interwoven skeleton of NIGMs is completely encapsulated by the polymeric gel matrix with an interpenetrated architecture.

## Excellent $CO_2$ separation performance of the NIGMs

To manifest the architecture advantage of NIGMs in gas separation, we first investigated the utility of CNT-interwoven skeleton by comparing $CO_2/N_2$ separation performance of NIGMs with nascent gel membranes (GMs), in which their gel layer thickness is fixed at about 4 μm. As shown in Fig. 3a, b, the NIGMs exhibit a boosted $CO_2$ permeance of 211.0 GPU, which is 16.6 times higher than that of GMs with a low $CO_2$ permeance of 12.7 GPU. Meanwhile, the NIGMs deliver a $CO_2/N_2$ selectivity value of 151, which surpasses the GMs by a factor of 3.9 (Fig. 3c, d). This concurrent boost in selectivity and permeance of NIGMs mainly originates from the function fusion of CNT-interwoven skeleton and defect-free $CO_2$-philic gel. On the one hand, CNT-interwoven skeleton can not only serve as ultrafast continuous transport pathways to boost $CO_2$ permeance, but also disrupt the stacking of polymer chains within gel networks for leading to heightened mobility of polymer chains and increased free volume for facilitating $CO_2$ diffusion and permeance (Supplementary Figs. 9 and 10). On the other hand, the in-situ formed $CO_2$-philic gel matrix has decent interfacial compatibility with CNT skeleton for avoiding non-selective defects, meanwhile harnessing its abundant polar ethylene oxide units to facilitate $CO_2$ solubility over $N_2$ for attaining enhanced selectivity.

The architecture-rendered excellent $CO_2$ separation performance of NIGMs can be also underpinned by using a pure gel layer-coated on the surface of NIGMs as control. Figure 3a, b depicts that the pure gel covered-NIGMs exhibit a bilayer architecture configuration, with a 1.2 μm-thick gel layer overlaying the NIGMs. Intriguingly, $CO_2$ permeance of this gel covered-NIGMs rapidly drops to 26.3 GPU, which is merely 12.3% of NIGMs with $CO_2$ permeance of 211.0 GPU. Analogously, $CO_2/N_2$ selectivity of the gel-coated NIGMs also diminishes to 84, representing only 55.6% of that observed for NIGMs with $CO_2/N_2$ selectivity of 151. It suggests that 3D-interconnected continuous transportation skeleton throughout the gel matrix is crucial for strengthening $CO_2$ separation performance (Fig. 3c). We further examined the impact of different stacking densities of CNT-interwoven skeleton on the $CO_2$ separation performance (Supplementary Fig. 11). When the CNT packing density is insufficient, the gel precursor is prone to penetrate into the underneath porous membrane, subsequently yielding a pure gel layer underneath the CNT-interwoven skeleton during photothermal-triggered gelation process. This gel architecture also results in a rapid reduction of $CO_2$ permeance (Supplementary Fig. 12). Conversely, an excessively high packing density of CNT-interwoven skeleton would result in an increased gel layer thickness of NIGMs, thereby augmenting the transport resistance and

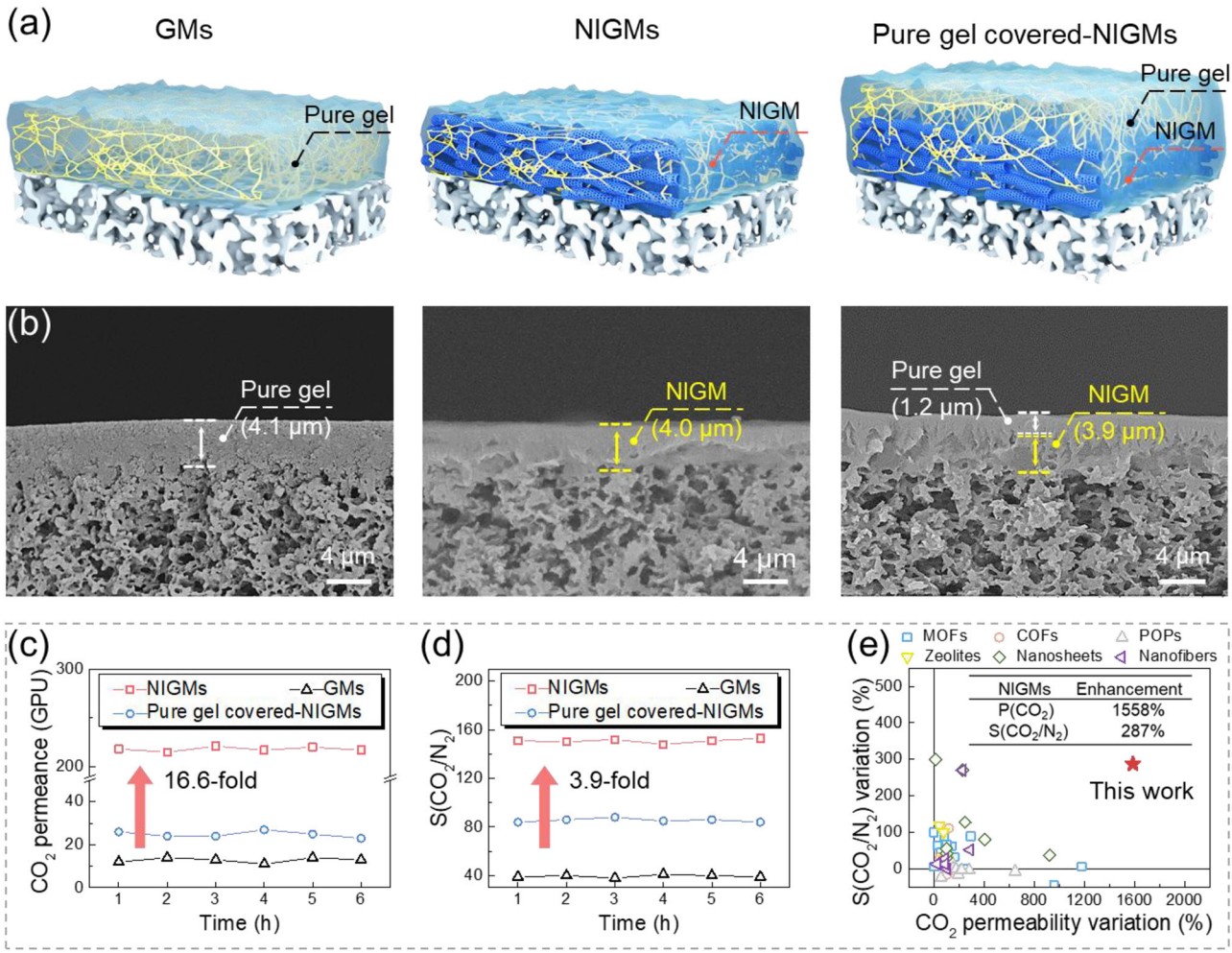

**Fig. 3 | Boosted permeance and selectivity of the NIGMs for $CO_2$ separation. a** Schematic illustration of GMs, NIGMs and pure gel covered NIGMs. **b** Cross-sectional images of GMs, NIGMs and pure gel covered NIGMs and their associated gel layer thickness. **c** $CO_2$ permeance and **d** $CO_2/N_2$ selectivity of GMs, NIGMs and pure gel covered NIGMs. **e** Comparison of the NIGMs and reported MMMs in terms of the improvement of $CO_2$ permeability and $CO_2/N_2$ selectivity.

ultimately reducing $CO_2$ permeance. For optimization, the NIGMs have a CNT packing density of 0.19 mg cm$^{-2}$ to achieve perfectional $CO_2$ separation performance.

The architecture merit of the NIGMs in $CO_2$ separation can be further reflected by comparing with the previously-reported MMMs. The majority of MMMs exhibit an unsatisfactory $CO_2$ permeance in the range of 1–100 GPU, accompanied with a $CO_2/N_2$ selectivity between 10 and 100. Although a minority of emerging MMMs containing advanced nanomaterials exhibit better $CO_2$ permeance than our NIGMs, they often suffer from unsatisfactory selectivity (20–60), far away from our NIGMs. Consequently, NIGMs occupy a favorable position in terms of both $CO_2$ permeance (211.0 GPU) and $CO_2/N_2$ selectivity (151) among the majority of MMMs. Further inspecting their performance enhancement after the introduction of nanomaterials, we found that the NIGMs exhibit a remarkable 1558% enhancement in $CO_2$ permeability and a notable 287% increase in $CO_2/N_2$ selectivity in comparison to GMs (Fig. 3e). In contrast, few MMMs exhibit a con-current boost in selectivity and permeability, in regardless of the types of nanomaterials employed, including MOFs, COFs, POPs, zeolites, nanosheets and nanofibers (Supplementary Table 1). Normally, due to the introduction of some advanced nanomaterials, the MMMs usually exhibit a significant increase in $CO_2$ permeability with a value of 50–400%, yet undergoing a decrease in $CO_2/N_2$ selectivity. This may be attributed to the formation of non-selective interfacial defects between nanomaterials and polymeric matrix. Therefore, our

nanofiber-interwoven skeleton within polymeric gel matrix holds great potential in improving permeance and selectivity for $CO_2$ separation simultaneously.

More impressive results can be obtained when examining $CO_2$ separation performance of NIGMs with various architectures of CNT-interwoven skeleton. To uncover the architecture impact of CNT-interwoven skeleton within NIGMs on gas separation perfor-mance, a series of NIGMs with various CNT-interwoven skeletons were synthesized utilizing different average lengths of CNT from $13.1 \pm 2.8\,\mu m$ (long CNT) to $6.8 \pm 1.5\,\mu m$ (medium CNT) and $1.4 \pm 0.3\,\mu m$ (short CNT) (Supplementary Fig. 13). Figure 4a shows that these three kinds of NIGMs, termed L-NIGMs, M-NIGMs, and S-NIGMs, have a uniform thickness of approximately 4 μm as well as similar nanomaterial loading content of about 60.3–66.7% (Sup-plementary Fig. 14). As expected, these NIGMs display distinct architectures of both CNT-interwoven skeleton and polymeric gel matrix. As the length of CNT decreases, the CNT-interwoven skele-ton undergoes a configuration transition from loose to dense, resulting in a notable reduction in the volume fraction of gel-rich region that normally is not beneficial for gas permeance. For example, L-NIGMs exhibit a loose CNT-interwoven skeleton, with the volume fraction of gel-rich region reaching up to 46.4%. In striking contrast, S-NIGMs comprise of a dominant volume fraction of CNT-interpenetrated gel region and a low volume fraction of gel-rich region below 1.0%, suggesting a highly uniform and intimate

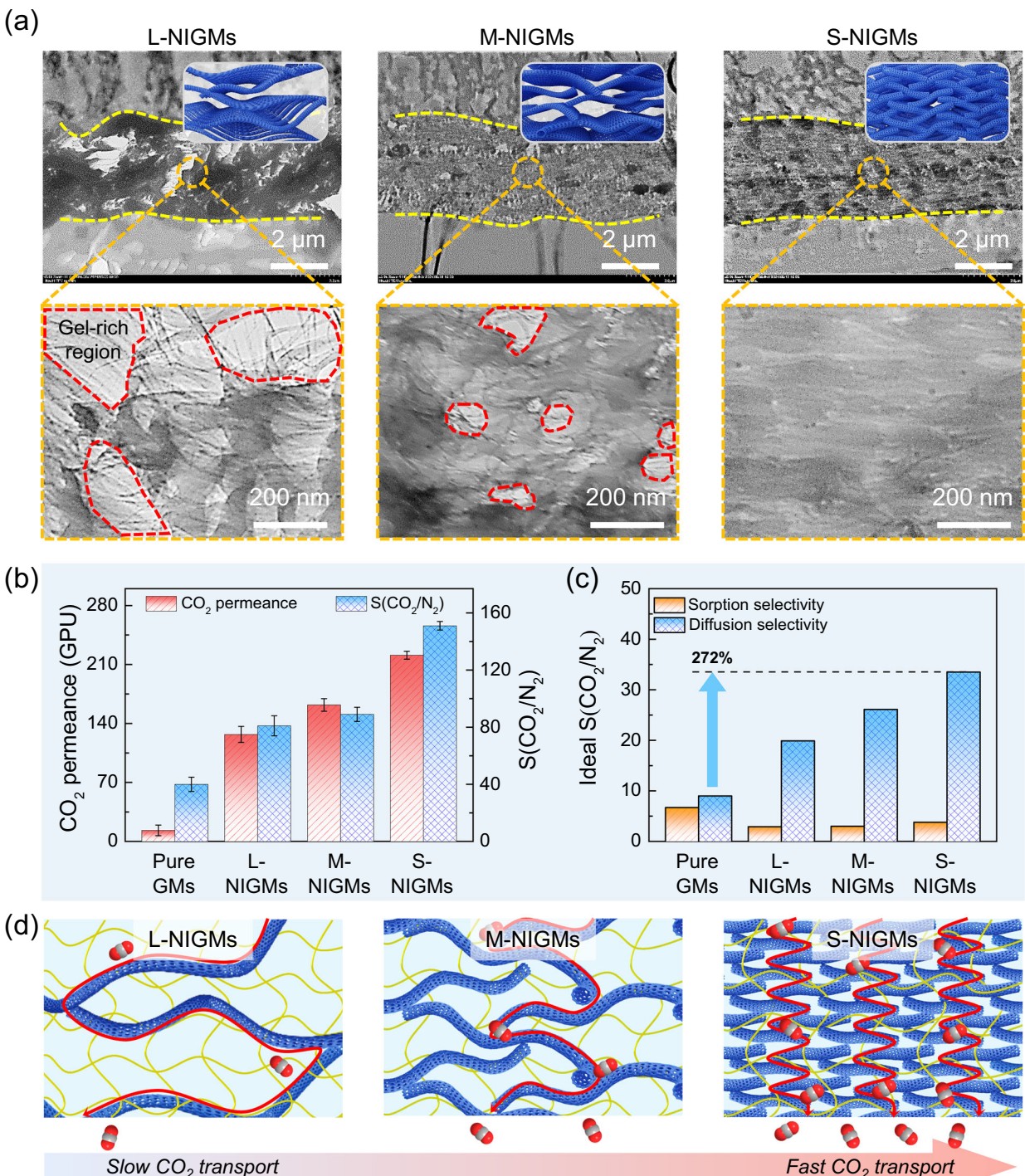

**Fig. 4 | CO₂ transport and separation mechanism of NIGMs with various CNT-interwoven skeletons. a** TEM images of various NIGMs (L-NIGMs, M-NIGMs and S-NIGMs), which feature distinct configurations of CNT-interwoven skeletons made of long CNT, medium CNT and short CNT, respectively. The insets are the according schematic illustration of various CNT-interwoven skeletons: long CNT, medium CNT and short CNT. **b** CO₂ permeance and CO₂/N₂ selectivity of various NIGMs (L-NIGMs, M-NIGMs and S-NIGMs) and GMs. Data presented as mean ± SD, $n = 5$. **c** Ideal sorption selectivity and diffusion selectivity of CO₂/N₂ for various NIGMs (L-NIGMs, M-NIGMs and S-NIGMs) and GMs. **d** Schematic illustration of CO₂ transport and separation mechanism through various NIGMs with distinct CNT-interwoven skeletons.

stacking of CNT-interwoven skeleton within polymeric gel matrix, which fosters abundant continuous transport channels for ultrafast CO₂ permeance. The gas separation performance of these NIGMs is found to be strongly correlated with the CNT-interwoven skeleton configuration (Fig. 4b). Specifically, L-NIGMs manifest the lowest CO₂ permeance of 127.6 GPU, while M-NIGMs exhibit an intermediate value of 162.4 GPU. Remarkably, S-NIGMs demonstrate superior performance, achieving a CO₂ permeance of 211.0 GPU. Moreover, the CO₂/N₂ selectivity of these NIGMs follows a similar increase trend from 81 to 151. Note that, all the NIGMs are equipped with superior CO₂ permeance and selectivity, far surpassing pure GMs and conventional MMMs.

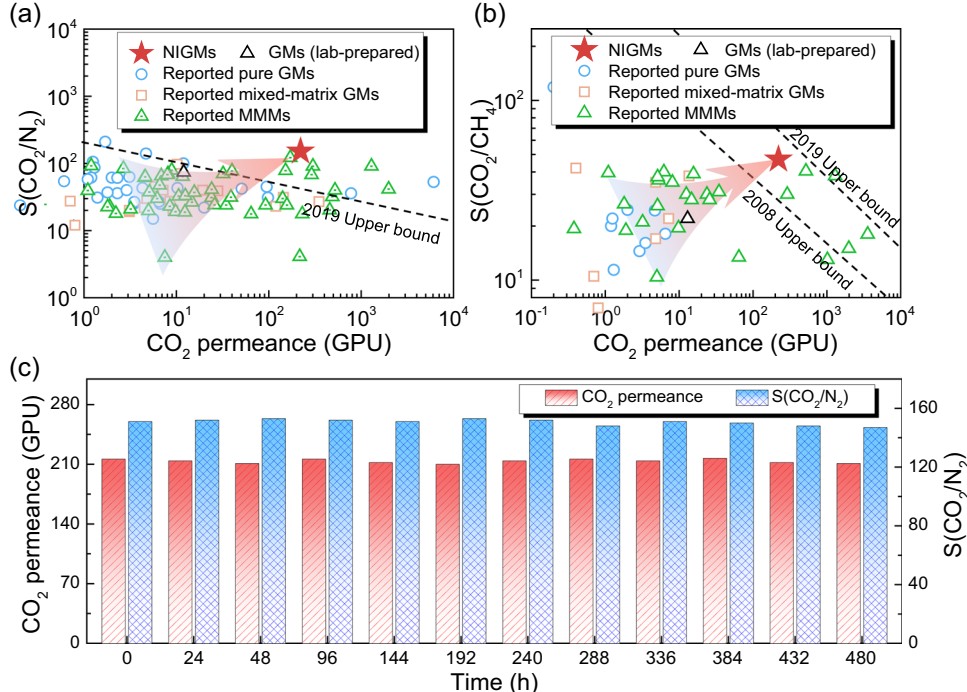

**Fig. 5 | Excellent and durable CO$_2$ separation performance of NIGMs.**
**a** Comparative evaluation of NIGMs against other GMs and reported conventional MMMs in terms of CO$_2$ permeance and CO$_2$/N$_2$ selectivity (The thickness of NIGMs is 4.0 μm). **b** Comparative evaluation of NIGMs against other GMs and reported conventional MMMs of CO$_2$ permeance and CO$_2$/CH$_4$ selectivity (The thickness of NIGMs is 4.0 μm). **c** Permeance and selectivity variation of NIGMs under long-term operation, exhibiting durable long-term service stability of NIGMs.

To gain more insights into CO$_2$ transport behavior within these distinct architectures of CNT-interwoven skeletons in NIGMs, we measured the solubility and diffusivity of CO$_2$ and N$_2$ in various NIGMs and nascent GMs, from which their ideal CO$_2$/N$_2$ sorption selectivity and diffusion selectivity can be calculated (Fig. 4c and Supplementary Figs. 15 and 16). As the volume fraction of CNT-interwoven gel region increases, the CO$_2$ diffusion coefficient and diffusion selectivity of NIGMs are observed to consistently rise in comparison to GMs. Remarkably, the diffusion selectivity of S-NIGMs reaches a remarkable value of 33.5, representing a 272% enhancement over that of GMs. In contrast, the sorption selectivity of our NIGMs is only 2.9–3.8, which is 8.8-fold lower than their diffusion selectivity. To further reveal CO$_2$ transport mechanism, we calculated fractional free volume for the NIGMs and pure GMs via molecular dynamics simulation. The results show that pure GMs have a fractional free volume of 6.9%, contrasting with the higher fractional free volume of 13.9% for the NIGMs (Supplementary Fig. 17). In addition to the fractional free volume, we also calculated diffusion coefficients of CO$_2$ nearby PEG matrix and nearby CNT within NIGMs (Supplementary Fig. 18). Significantly, the inclusion of the CNT triggers a substantial elevation in gas diffusivity, escalating from $5.23 \times 10^{-8}$ cm$^2$ s$^{-1}$ for CO$_2$ nearby PEG matrix to an impressive $1.16 \times 10^{-5}$ cm$^2$ s$^{-1}$ for CO$_2$ nearby CNT within NIGMs (Supplementary Movie 1). Above simulation results harmonize seamlessly with outcomes derived from measured diffusivity. These findings indicate that CO$_2$ transport and separation mechanism within NIGMs follows a classic solution-diffusion model and is heavily dictated by the diffusion process. This diffusion-dominated mechanism mainly originates from the densely-packed CNT-interwoven skeleton as an interconnected continuous transport network for effectively shortening CO$_2$ transport pathways (Fig. 4d). Taken together, by meticulously orchestrating the distribution and stacking configuration of CNT-interwoven skeleton, the architecture of NIGMs can be optimized for attaining exceptional CO$_2$ separation performance, with great implication for developing high-performance CO$_2$ separation membranes.

Moving from the CNT-interwoven skeleton of NIGMs, we also investigated the impact of their polymeric gel matrix on CO$_2$ separation performance. By controlling the conversion of PEG monomer, the NIGMs can accommodate some unreacted PEG monomer as function additive to enhance the mobility of polymer chains as well as act as CO$_2$ carriers within polymeric gel network, fostering a more facile diffusion of CO$_2$. As the fraction of unreacted monomers increases to 71%, CO$_2$ permeance of NIGMs exhibits a substantial increase from 89.2 GPU to 211.0 GPU, while maintaining a CO$_2$/N$_2$ selectivity exceeding 116 (Supplementary Fig. 19). Further increasing the fraction of unreacted monomers, the as-fabricated NIGMs would encounter a significant deterioration in structural stability, leading to an unstable CO$_2$ separation performance. In addition to the unreacted monomers within NIGMs, the CO$_2$ separation performance of NIGMs can be also regulated by varying molecular weights of PEG monomers (Supplementary Fig. 20). As the molecular weight of the PEG monomers increases from 200 to 1000, the CO$_2$ permeance of NIGMs drops sharply from 211.0 GPU to 26.2 GPU. This is because the polymer chains of gel matrix fabricated by higher molecular weight of PEG monomer are prone to form intramolecular and intermolecular interactions and even occur local crystallization, which impedes CO$_2$ diffusion and consequently reduces CO$_2$ permeance. Therefore, designing and optimizing chemical component of polymeric gel matrix offers an easy yet powerful toolkit to enhance CO$_2$ separation performance of NIGMs.

To highlight the application potential in CO$_2$ separation, the NIGMs are challenged with previously reported GMs and reported MMMs. Figure 5a shows that our NIGMs surpass the widely accepted 2019 upper bound limit for CO$_2$/N$_2$ separation (Supplementary Table 2), which stands in contrast to that most reported GMs and MMMs are below 2019 upper bound limit. It indicates that the NIGMs hold giant potential in the application of carbon capture materials. Moreover, the excellent gas separation performance of our NIGMs can be successfully translated into CO$_2$/CH$_4$ separation for natural gas

purification (Fig. 5b and Supplementary Table 3). The $CO_2$ permeance of our NIGMs is found to be significantly improved, reaching up to 211.0 GPU, while their $CO_2/CH_4$ selectivity has been determined to be 47. By comparing with other reported GMs and MMMs, our NIGMs manifest great competitiveness in $CO_2$ permeance and $CO_2/CH_4$ selectivity that exceed the 2008 upper bound and approach the 2019 upper bound limit, positioning them among the state-of-the-art MMMs. Notably, compared with GMs, our NIGMs demonstrate a significant enhancement in $CO_2$ permeance and $CO_2/CH_4$ selectivity by ~1783% and ~230% respectively, underpinning the merits of our NIGMs in architecture design. To further underscore the inherent permeability advantages of our NIGMs, we conducted a rigorous comparative analysis of $CO_2$ permeability against state-of-the-art membranes. The $CO_2$ permeability of our NIGMs is 844.0 Barrer, with $CO_2/N_2$ selectivity surpassing the 2019 Robeson upper bound and $CO_2/CH_4$ selectivity exceeding the 2008 Robeson upper bound (Supplementary Fig. 21). The remarkable performance of NIGMs can be attributed to the function fusion of the densely-packed CNT-interwoven skeleton and interpenetrated defect-free gel matrix.

In paralleled with the excellent $CO_2$ separation performance, the architecture fusion of CNT-interwoven skeleton and interpenetrated polymeric gel matrix can be also employed to attain excellent stability under long-term operation. Normally, conventionally polymer membranes face many issues such as physical aging and $CO_2$ plasticization, leading to a significant decrease in long-term separation performance. Nevertheless, our NIGMs are capable of leveraging mechanical strength of CNT-interwoven skeleton for functioning as a rigid skeleton in the interior to stabilize flexible polymeric gel matrix. At the microscopic scale, there exists strong topological chain entanglement and multiple non-covalent interactions between CNT-interwoven skeleton and polymeric gel network to create robust architecture (Supplementary Fig. 22). These results in the formation of a robust multidimensional framework that collectively grants exceptional service stability of NIGMs. Figure 5c depicts that $CO_2$ permeance of NIGMs was maintained at 212.4 GPU after 480 h, exhibiting a mere 1.7% decrease in comparison to the initial state. Such a slight fluctuation is markedly lower than that of previous advanced $CO_2$ separation membranes with a 10–20% permeance variation upon the same separation time[28]. Concurrently, the $CO_2/N_2$ selectivity of the NIGMs is persistently superior to 148, indicating that the performance of NIGMs surpass the 2019 upper bound in terms of selectivity and permeance even after long-term operation. This evidence further underscores the exceptional stability and durability of our NIGMs under long-term service.

In real-world $CO_2$ capture applications, some harsh conditions such as high humidity and temperature environments are often encountered. We further investigated the influence of the humidity and temperature on the $CO_2$ separation performance of our NIGMs. NIGMs display enhanced $CO_2$ permeance but diminished selectivity in damp environments. As the relative humidity (RH) value is elevated from 0% to 70%, the $CO_2$ permeance of the NIGMs increases by 8.5%, attaining 229.3 GPU, while its $CO_2/N_2$ selectivity decreases by 45.6% to only 82.5 (Supplementary Fig. 23). It is noteworthy that our NIGMs demonstrate dynamic responsiveness to ambient humidity cycling, with permeance and selectivity gradually restoring to initial values upon re-exposure to dry environments (RH = 0%). In analogy with the humidity, the $CO_2$ permeance of NIGMs increases monotonically with temperature, reaching 297.4 GPU at 40 °C and further escalating to 406.1 GPU at 60 °C (Supplementary Fig. 24). However, $CO_2/N_2$ selectivity of NIGMs displays a decrease trend as temperature increases, from 151.2 at 25 °C to 73.1 at 60 °C. This can be attributed to the decreased $CO_2$ solubility within the PEG matrix at higher temperatures, which decreases both $CO_2/N_2$ solubility selectivity and overall separation selectivity. To recap, our NIGMs still possess good $CO_2$ separation performance even under high humidity and temperature environments.

## Discussion

In this study, a kind of nanofiber-interwoven gel membranes (NIGMs) that exhibit remarkable $CO_2$ separation performance are designed and fabricated through a facile photothermal-triggered in situ gelation approach. Distinct from conventional MMMs fabricated by solvent evaporation method, our photothermal-triggered NIGMs feature preferential interfacial compatibility between nanofiber scaffold and polymeric gel matrix even under a high nanomaterial loading of 66.7% without the occurrence of non-selective defects. With this method, the NIGMs can leverage the architecture and function fusion of a CNT-interwoven continuous transportation skeleton and an in-situ formed interpenetrated $CO_2$-philic gel matrix for boosting selectivity while maintaining high $CO_2$ permeance. By optimizing the distribution and stacking density of CNT-interwoven skeleton, the NIGMs give exceptional $CO_2/N_2$ permeance of 211.0 GPU increased by 1558% and an ultrahigh selectivity of up to 151 increased by 287%, compared to the polymeric gel counterpart.

From a broader perspective, the design concept of nanofiber-interwoven skeleton disclosed herein, can parasitize other functional separation layers, pushing the practical application of advanced MMMs into broader application scenarios ranging from gas separation and water purification. Although recent studies have employed nanofibers as facilitated transport pathways[29–32]. Yet, these membranes frequently suffer from unsatisfactory performance, due to the inefficient and disconnected bulky transport channels as well as inferior interfacial compatibility between nanofibers and polymeric matrix. The original architecture presented in our work leverages densely-stacked nanofiber-interwoven skeleton as an interconnected continuous transport network for boosting permeance, while in-situ formed gel matrix has decent interfacial compatibility with nanofiber-interwoven skeleton to construct defect-free membranes even under high loading nanomaterials. Our findings offer a direction for designing high-performance MMMs.

## Methods
### Materials
Nylon membranes (0.2 μm) and anodic alumina oxide membranes (0.2 μm) were procured from GE healthcare Ltd (UK). Iron(III) chloride hexahydrate ($FeCl_3 \cdot 6H_2O$) were supplied by Sigma-Aldrich (USA). Other chemicals, including ammonium persulfate (($NH_4$)$_2S_2O_8$), lithium nitrate ($LiNO_3$), poly(ethylene glycol) diacrylate (PEGDA, M$w$ ~ 200 g mol$^{-1}$), PEGDA (M$w$ ~ 600 g mol$^{-1}$) and PEGDA (M$w$ ~ 1000 g mol$^{-1}$) were sourced from Aladdin Chemical Co., Ltd (China). Gases such as carbon dioxide ($CO_2$), nitrogen ($N_2$) and methane ($CH_4$) were provided by Hangzhou Jingong Special Gas Co., Ltd (China). The purified water used in this study was generated in our laboratory using an ELGA LabWater system (France). CNT nanofibers (single-walled carbon nanotube dispersions) were bought from Chengdu Organic Chemicals Co. Ltd. Prior to experiments, the commercial CNT dispersions underwent centrifugation at 1118 × $g$ for a period of 90 min. The precipitate obtained from the aforementioned experiment was then dispersed in water using ultrasonication to prepare dispersions of long CNT and the supernatant was subjected to centrifugation at 4472 × $g$ for an additional 90 min. The resultant precipitate was dispersed in water through ultrasonication, yielding dispersions of medium CNT. The remaining supernatant constituted the dispersions of short CNT.

### Fabrication of nanofiber-interwoven gel membranes (NIGMs)
The NIGMs were fabricated using a facile photothermal-triggered in situ gelation strategy and were implemented in a photothermal confined reactor made of densely-stacked CNT-based nanofibers. Specifically, the photothermal confined reactor was first constructed by steering the assembly and stacking of CNTs (0.19 mg cm$^{-2}$) into a nanofiber-interwoven skeleton onto the surface of nylon membranes

through vacuum filtration. Subsequently, the as-prepared CNT-interwoven photothermal confined reactor was infused with gel precursor. The gel precursors, prepared in advance, were a dispersion of PEGDA (9.5 g), and thermal initiator, $(NH_4)_2S_2O_8$ (95 mg) in water (0.5 g). Finally, the CNT-interwoven photothermal confined reactor was exposed to 2-suns irradiation for a predetermined duration, leading to the formation of NIGMs.

## Evaluation of monomer conversion ratio of NIGMs

Monomer conversion ratio of the NIGMs was quantitatively analyzed via the employment of internal standard calibration and a laser Raman spectrometer (inVia Reflex, Renishaw plc, UK). Initially, a special gel precursors with internal standards were prepared by dispersing PEGDA (9.5 g), internal standards ($LiNO_3$, 0.5 g), and thermal initiator ($(NH_4)_2S_2O_8$, 95 mg) in water (0.5 g). Subsequently, densely-stacked CNT-based nanofibers were perfused with the prepared gel precursors. These membranes were then subjected to 2-suns irradiation for a predetermined duration (*t*). Upon completion of the reaction, the membranes were promptly immersed in a 0.5 M $FeCl_3$ solution (1 mL) to halt the polymerization process. The composite gel membranes were uniformly dispersed in above solution using ultrasonic treatment, and the resulting mixture was filtered to eliminate insoluble matter. The filtered solution was then analyzed using the laser Raman spectrometer to determine the peak areas at 1047 cm$^{-1}$ (associated with $LiNO_3$) and 1640 cm$^{-1}$ (indicative of the carbon-carbon double bond of unreacted PEG monomers). The monomer conversion ratio (Conversion ratio$_t$, %) of the NIGMs was calculated using the following Eq. (1):

$$Conversion\ ratio_t = \frac{A_{C=C,\ t0}/A_{LiNO3,\ t0} - A_{C=C,\ t}/A_{LiNO3,\ t}}{A_{C=C,\ t0}/A_{LiNO3,\ t0}} \times 100\% \quad (1)$$

where $A_{C=C,t}$ and $A_{C=C,t0}$ represent the peak areas of the carbon-carbon double bond of the unreacted monomers at a given reaction time (*t*) and initially ($t_O$), respectively. Similarly, $A_{LiNO3,t0}$ and $A_{LiNO3,t}$ denote the peak areas of $LiNO_3$ initially and after a specific reaction time, respectively.

## Measurement of gas separation performance of NIGMs

Gas permeance tests were conducted at room temperature (298 K) using a self-built permeation device employing the constant pressure-variable volume method (Supplementary Fig. 25). Specifically, the membranes were first fixed in a membrane module, which was placed inside a temperature-controlled oven to maintain isothermal conditions. Before experiment, the whole pipeline of the permeation device was purged with the test gas for at least 1 min. During experiment, the upstream pressure of the module was set at 151.0 cmHg, while the downstream pressure was maintained at 76.0 cmHg (atmospheric conditions). The gases were tested in sequential order: $N_2$, $CH_4$ and $CO_2$. Downstream gas flow rates were monitored with a digital bubble flowmeter. After the system reached steady-state, all gas permeation measurements were performed more than three times, and the reported data includes the average values along with the standard deviation. The evaluation of the gas separation performance of the NIGMs involved the quantification of gas permeance (*P*, GPU), gas permeability (*G*, Barrer) and ideal selectivity ($S_{CO_2/X}$).

The gas permeance was calculated by:

$$P = \frac{V \times T_0 \times 10^6}{\triangle t \times S \times \triangle p \times T} \quad (2)$$

where *V* represents the volume of gas permeating through the membrane (cm$^3$), $\triangle t$ denotes the permeation time (s), *S* is the effective separation area of the membranes (2.2 cm$^2$), $\triangle p$ indicates the pressure

difference between the upstream and the downstream of the membranes (75.0 cmHg), $T_0$ is 273.15 K and *T* is the test temperature (298.15 K).

The gas permeability was calculated by:

$$G = P \times L \quad (3)$$

where *L* is the membrane thickness (μm).

The ideal selectivity was determined using the following equation:

$$S_{CO_2/X} = \frac{P_{CO_2}}{P_X} \quad (4)$$

where $P_{CO_2}$ represents the $CO_2$ permeance of NIGMs (GPU), and *X* refers to the $N_2$ or $CH_4$.

## Other characterization

The optical absorption spectra of the membranes were deduced by following equation:

$$A = (1 - R - T) \times 100\% \quad (5)$$

where *R* and *T* represent reflection and transmission, respectively. Both reflectance (*R*) and transmittance (*T*) spectra of the membranes were recorded using a spectrophotometer (UH4150, HITACHI, Japan) within the wavelength range of 200 to 2500 nm.

The dynamic surface temperature of the membranes during gelation process was monitored using an IR imaging device (FLIR ONE PRO, FLIR Systems Inc., USA). The free volume of the GMs and NIGMs was detected by X-ray diffractions (XRD, Bruker D8 Discover, Bruker, Germany) over an angular range of 0.5° to 50°. The CNT loading content of the NIGMs were quantified using thermogravimetric analysis (TGA, Q50, TA Instruments, USA) under $N_2$ atmosphere with a heating rate of 10 °C min$^{-1}$ from room temperature to 700 °C. The $CO_2$ and $N_2$ adsorption isotherm of NIGMs were measured by Micromeritics ASAP 2460 (USA) at 298 K.

## Data availability

The authors declare that all the data supporting the findings of this study are available within the article and Supplementary Information. All data are available from the corresponding author upon request.

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

## Acknowledgements

This work is financially supported by the National Natural Science Foundation of China (Grant no. 22375174 to C.Z.), the Natural Science Foundation of Zhejiang Province (Grant no. LZ24E030001 to C.Z.), and the Fundamental Research Funds for the Central Universities (Grant no. 226-2024-00027 to C.Z.). C.Z. also acknowledges gratefully research startup package from Zhejiang University.

## Author contributions

C. Zhang conceived and designed the overall study. H.-N. Li, Z.-Y. Sun and K. Man performed experiments and molecular simulation. H.-N. Li, C. Zhang and Z.-K. Xu drafted the manuscript. H.-N. Li and Z.-J. Yu designed and prepared the figures. All the authors contributed to the discussion, and data analysis and revised the manuscript.

## Competing interests

The authors declare no competing interests.
