## [Transparent Peer Review file · Nature Communications]

Nanofiber-interwoven gel membranes with tunable 3D-interconnected transport channels for efficient CO₂ separation

Corresponding Author: Professor Chao Zhang

Version 0:

Reviewer comments:

Reviewer #1

(Remarks to the Author)

In this manuscript, the authors present a novel architecture of nanofiber-interwoven gel membranes (NIGMs) consisting of CNT-interwoven interconnected ultrafast transport highways and highly-selective CO₂-philic gel matrix, enabling a prominent boost in both permeance and selectivity towards CO₂ separation. The subject matter and main findings of this study are interesting. Careful revision is necessary before consideration for publication. Specific comments on this manuscript are as follows:

1. The novelty of the work should be established in the "Main" section, as you mentioned in the abstract.
2. The CNT interwoven skeleton is mentioned several times in the manuscript as being used as a photothermal confinement reactor, and this needs to be mentioned in the abstract section.
3. Is there a test standard for CO₂ separation? How is the testing performed?
4. There are many SEM pictures in the manuscript and supporting materials, and the title shows that the interwoven network of nanofibers is highly essential, so it is suggested to provide more data on nanofibers.
5. What are the advantages of gel membranes over membranes in the traditional sense? It needs to be reflected in the manuscript.
6. There have been many recent publications on membranes and separation performance, and it is necessary to briefly describe what has been achieved in the literature and the gaps that the authors are trying to fill (10.1016/j.cej.2023.145403; 10.1016/j.matdes.2020.108591).

Reviewer #2

(Remarks to the Author)

In this study, nanofiber-interwoven gel membranes (NIGMs) were developed using a photothermal-triggered in-situ gelation approach, ensuring strong interfacial compatibility at high nanomaterial loadings. These membranes achieve an exceptional CO₂/N₂ permeance of 211 GPU and high selectivity of 151, surpassing the 2019 upper bound for CO₂ separation, marking a breakthrough in membrane design.

There are many ways to improve the compatibility between fillers and the matrix in MMMs, but the reported system is somewhat intriguing and innovative. However, I cannot recommend publication in its current form. I strongly suggest a thorough revision, including the removal of overly bold claims and a more attentive comparison of the results with existing literature. In my opinion, the paper could be resubmitted and reconsidered for publication after the authors address the following points:

- The authors refer to a 44-fold increase in permeance (page 5), which seems highly specific. Compared to what? Instead of broadly referring to "conventional polymeric membranes," they should provide specific examples and references to clarify the comparison.
- Since permeance depends on membrane thickness, it is crucial to state the membrane thickness used and what it is compared to when reporting GPU values. A CO₂ permeance of 211 GPU is relatively high, but whether it is truly exceptional depends on the benchmark used. A comparison with lower-permeability polymers may make the results seem extraordinary, but a more balanced evaluation is needed.
- I strongly advise the authors to remove over-emphatic statements such as "extraordinary CO₂ separation." Performance

evaluation depends on multiple parameters, and while the results are very good, calling them extraordinary is an overstatement. Even in Table S1 and Figure 5, some reported results are, at best, comparable to existing data.

- On page 8, the authors refer to “exceptional permeability,” stating that it is “16.6 times higher than conventional GMs with a permeance of 12.7 GPU.” Can they provide references and evidence to confirm that no other materials achieve similar results? This is another very bold statement.
- I agree with the statement that the reported results are very good, but a fairer comparison with other polymers and MMMs could be achieved by including permeability data in Robeson plots and reporting values in Barrer. While GPU is useful for industrial applications, Barrer is more appropriate for comparing materials, as it eliminates variations due to differences in membrane thickness.
- If the permeation mechanism is mainly based on gas diffusion and molecular sieving, I would expect the materials to exhibit high porosity. However, a key aspect missing from this study is porosity assessment using low-temperature isothermal adsorption (e.g., N₂ @ 77 K and/or CO₂ @ 273 K). This is particularly relevant since the authors claim the formation of a dense, defect-free layer, which may not be as porous as suggested.
- The last paragraph should be titled “Conclusions” rather than “Discussion.”

Reviewer #3

(Remarks to the Author)

This manuscript reports an innovative and highly impactful strategy for fabricating mixed matrix membranes (MMMs) with unprecedented CO₂ separation performance. The authors propose a bio-inspired “nanofiber-interwoven gel membrane” (NIGM) architecture that integrates a carbon nanotube (CNT) network and a CO₂-philic polymer gel matrix via a photothermal-triggered in-situ gelation process. The result is a defect-free membrane that achieves both exceptional CO₂ permeance (211 GPU) and CO₂/N₂ selectivity (151), outperforming the Robeson 2019 upper bound and most reported MMMs. The study is well-structured, systematically executed, and thoroughly benchmarked. It introduces a fabrication method of general utility and presents significant advancement in gas separation membrane design. I believe this work is suitable for publication in Nature Communications after addressing a few minor comments.

1. Phrases such as “cardiacmuscle-inspired ultrafast transport channels” may appear overly metaphorical for a materials science audience. I suggest rephrasing these in more conventional membrane science terms to improve scientific clarity.
2. Please briefly discuss the potential for scaling up the photothermal-triggered fabrication method. How feasible is sunlight-based gelation for industrial-scale membrane production? Could alternative irradiation sources be employed?
3. While the diffusion-dominated mechanism is well supported by data, inclusion of additional discussion on gas pathway modeling or free volume distributions (even qualitatively) would enhance mechanistic understanding.
4. Consider adding commentary on how these membranes might perform under varied humidity or temperature cycling, as this would be relevant for real-world CO₂ capture applications.

Version 1:

Reviewer comments:

Reviewer #1

(Remarks to the Author)

The authors have brought valuable corrections to give the submitted manuscript a much better shape. This work is now more accessible and fits more to the journal standard. The manuscript might be published then.

Reviewer #2

(Remarks to the Author)

In my opinion, the authors have made a good effort in addressing all the concerns raised by the reviewers, and I believe the paper could be accepted.

[Editor's Note: This reviewer was asked to evaluate the rebuttal comments to Reviewer #3 and provided the following additional comment]

I had already seen the comments of all reviewers and it seems acceptable to me. They put some effort in trying to address all the concerns

Point-to-point responses to reviewers' comments

Response to the comments by Reviewer 1

Comment 1: In this manuscript, the authors present a novel architecture of nanofiber-interwoven gel membranes (NIGMs) consisting of CNT-interwoven interconnected ultrafast transport highways and highly-selective CO₂-philic gel matrix, enabling a prominent boost in both permeance and selectivity towards CO₂ separation. The subject matter and main findings of this study are interesting. Careful revision is necessary before consideration for publication.

Response: We are pleased to see that the reviewer found our work interesting. The insightful comments raised by the reviewer help improve the merit of our paper.

Comment 2: The novelty of the work should be established in the “Main” section, as you mentioned in the abstract.

Response and revision: Following the reviewer’s suggestion, we have highlighted the novelty of our work in the “Main” section of the revised manuscript.

In **Page 3:** “The NIGMs are formulated by an innovative photothermal-triggered in-situ gelation method, in which the CNT-interwoven skeleton is harnessed as a photothermal confined reactor to create highly-crosslinked CO₂-philic gel exhibiting an interpenetrated architecture with CNT-interwoven skeleton. The in-situ formed CO₂-philic gel matrix has decent interfacial compatibility with CNT-interwoven skeleton without the occurrence of non-selective defects, meanwhile harnessing its abundant polar ethylene oxide units to facilitate CO₂ solubility over N₂ for attaining enhanced selectivity. Moreover, the distribution, stacking and configuration of interwoven nanofibers in our NIGMs can be easily regulated for creating 3D-interconnected continuous skeleton as ultrafast transport pathways to boost CO₂ permeance.”

Comment 3: The CNT interwoven skeleton is mentioned several times in the manuscript as being used as a photothermal confinement reactor, and this needs to be mentioned in the abstract section.

Response and revision: Following the reviewer’s suggestion, we have highlighted the photothermal confinement reactor of CNT-interwoven skeleton in the abstract section of the revised manuscript.

In **Page 1:** “CNT-interwoven skeleton: (1) serving as a photothermal confined reactor that rapidly triggers in-situ gelation of highly-selective CO₂-philic gel without phase separation-induced interfacial defects to construct defect-free and thickness-controllable NIGMs;”

Comment 4: Is there a test standard for CO₂ separation? How is the testing performed?

Response and revision: Among the various methods described in the literatures for measuring gas separation performance, three standardized methods are widely adopted: the variable volume, the variable pressure, and the variable concentration methods (*J. Appl. Polym. Sci.*, **1963**, 7, 2035-2051). In our work, the gas separation performance of NIGMs was measured using a self-built permeation device (Supplementary Fig. 25) employing the variable volume method, which have been widely used in many previous reports (*Nat. Commun.*, **2018**, 9, 990; *Sci. Adv.*, **2018**, 4, eaau1698; *Science*, **2013**, 342, 91-95).

Specifically, the membranes were first fixed in a membrane module, which was placed inside a temperature-controlled oven to maintain isothermal conditions (25 °C). Before experiment, the whole pipeline of the permeation device was purged with the test gas for at

least 1 minute. During experiment, the upstream pressure of the module was set at 151.0 cmHg, while the downstream pressure was maintained at 76.0 cmHg (atmospheric conditions). The gases were tested in sequential order: N₂, CH₄ and CO₂. Downstream gas flow rates were monitored with a digital bubble flowmeter. After the system reached steady-state, all gas permeation measurements were performed more than three times, and the reported data includes the average values along with the standard deviation. The evaluation of the gas separation performance of the NIGMs involved the quantification of gas permeance (P , GPU) and ideal selectivity ($S_{CO_2/X}$).

The gas permeance was calculated by: $P = \frac{V \times T_0 \times 10^6}{\Delta t \times S \times \Delta p \times T}$, where V represents the volume of gas permeating through the membrane (cm³), Δt denotes the permeation time (s), S is the effective separation area of the membranes (2.2 cm²), Δp indicates the pressure difference between the upstream and the downstream of the membranes (75.0 cmHg), T_0 is 273.15 K and T is the test temperature (298.15 K).

The ideal selectivity was determined using the following equation: $S_{CO_2/X} = \frac{P_{CO_2}}{P_X}$, where P_{CO_2} represents the CO₂ permeance of NIGMs (GPU), and X refers to the N₂ or CH₄.

In response to the reviewer's suggestions, a schematic diagram of gas permeation test has been added in the revised manuscript.

Supplementary Fig. 25 Schematic diagram of gas permeation test.

Comment 5: There are many SEM pictures in the manuscript and supporting materials, and the title shows that the interwoven network of nanofibers is highly essential, so it is suggested to provide more data on nanofibers.

Response and revision: Following the reviewer's suggestion, we have replaced the previous SEM pictures in Fig. 2 with AFM images and TEM image of CNT-interwoven skeleton within NIGMs. The AFM image shows that there is the interwoven network of nanofibers with high stacking density to generate 3D-interconnected architecture (Fig. 2d). Following the photothermal-triggered in-situ gelation process, the CNT-interwoven skeleton is uniformly wrapped with a dense polymeric gel matrix layer, without observable voids or defects (Fig. 2e). The surface morphology of the CNT-interwoven skeleton within NIGMs can be also clearly observed in the AFM and TEM image, indicating that the highly-crosslinked CO₂-philic gel is interpenetrated with the CNT-interwoven skeleton.

We have included and discussed these results in the revised manuscript.

Fig. 2 Fabrication and structure characterization of NIGMs. (a) Schematic illustration of the fabrication process of NIGMs by a facile photothermal-triggered in-situ gelation method. (b) Infrared images of the CNT-interwoven photothermal confined reactor as a function of sunlight irradiation time. (c) Dynamic curve of temperature and monomer conversion ratio over reaction time for the fabrication of NIGMs via photothermal-triggered in-situ gelation method. (d, e) AFM images and corresponding SEM images of surface morphologies of the densely-packed CNT-interwoven skeleton and the associated NIGMs. (f) Cross-sectional TEM and SEM images of the NIGMs. The CNT-interwoven skeleton of NIGMs is completely encapsulated by the polymeric gel matrix with an interpenetrated architecture.

Comment 6: What are the advantages of gel membranes over membranes in the traditional sense? It needs to be reflected in the manuscript.

Response and revision: Thank the reviewer for pointing out our unclear description about the advantages of gel membranes over traditional counterparts. In response to the reviewer's suggestions, we have highlighted this in the manuscript as follows:

In **Page 5**: “Moreover, the polymeric skeleton of our NIGMs is also different from traditional polymeric membranes that suffer from neither dense polymer chain packing nor limited CO₂ transport carriers. The NIGMs not only are made of flexible crosslinked gel networks with good chain mobility for enlarged fractional free volume, but also consist of numerous functional carriers to accelerate CO₂ transport, both of which lead to preferential CO₂ solution and diffusion over N₂ for attaining enhanced selectivity.”

Comment 7: There have been many recent publications on membranes and separation performance, and it is necessary to briefly describe what has been achieved in the literature and the gaps that the authors are trying to fill (10.1016/j.cej.2023.145403; 10.1016/j.matdes.2020.108591).

Response and revision: According to the reviewer's suggestion, we found out these literatures and carefully read them. These documents are helpful to understand the current

research progress regarding the application of nanofiber-based thin-film-composite membranes in the fields of nanofiltration and forward osmosis. We have added and discussed these literatures the revised manuscript.

In **Page 19**: “From a broader perspective, the design concept of nanofiber-interwoven skeleton disclosed herein, can push the development and applications of advanced MMMs into broader application scenarios ranging from gas separation to water purification. Indeed, many previous studies have employed nanofibers as facilitated transport pathways²⁹⁻³², however, these nanofiber-based MMMs suffer from unsatisfactory separation performance, due to the low-loading content and discrete distribution of nanofibers within polymeric matrix, as frequently encountered in the conventional solvent evaporation method of MMMs. Alternatively, the design architecture presented in our work leverages densely-stacked nanofiber-interwoven skeleton as an interconnected continuous transport network for boosting permeance, while in-situ formed gel matrix has decent interfacial compatibility with nanofiber-interwoven skeleton to construct defect-free membranes even under high loading nanomaterials. Our findings bridge the gap between design principle and performance of nanofiber-interwoven MMMs. ”

References

29. Miao, L. et al. Asymmetric forward osmosis membranes from *p*-aramid nanofibers. *Mater. Des.* **191**, 108591 (2020).
30. Li, S. et al. Fabrication of bamboo cellulose-based nanofiltration membrane for water purification by cross-linking sodium alginate and carboxymethyl cellulose and its dynamics simulation. *Chem. Eng. J.* **473**, 145403 (2023).
31. Zheng, W. et al. Continuous facilitated transport pathway constructed by in-situ interlinkage of mobile aniline with fixed carriers distributing along nanofibers for carbon capture. *J. Membr. Sci.* **697**, 122500 (2024).
32. Li, L. et al. Constructing the interactive “neuron-inspired” transport network in membranes for efficient CO₂/CH₄ separation. *J. Membr. Sci.* **720**, 123775 (2025).

Response to the comments by Reviewer 2

Comment 1: In this study, nanofiber-interwoven gel membranes (NIGMs) were developed using a photothermal-triggered in-situ gelation approach, ensuring strong interfacial compatibility at high nanomaterial loadings. These membranes achieve an exceptional CO₂/N₂ permeance of 211 GPU and high selectivity of 151, surpassing the 2019 upper bound for CO₂ separation, marking a breakthrough in membrane design.

There are many ways to improve the compatibility between fillers and the matrix in MMMs, but the reported system is somewhat intriguing and innovative. However, I cannot recommend publication in its current form. I strongly suggest a thorough revision, including the removal of overly bold claims and a more attentive comparison of the results with existing literature. In my opinion, the paper could be resubmitted and reconsidered for publication after the authors address the following points:

Response: We thank the reviewer for his/her comment. Following the reviewer’s suggestions, we have carefully examined and deleted the hyperbole and unscientific language, and substantially revised the manuscript.

As stated by the reviewer, many approaches such as exploiting new nanomaterials and surface modification of nanomaterials have been reported to improve the compatibility between fillers and polymer matrix in MMMs. However, our work presents a photothermal confined in-situ gelation strategy that can eliminate phase separation-induced interfacial

defects to construct defect-free NIGMs even under high-loading nanomaterials. Moreover, our strategy is also generic and can be extended for other materials such as nanofibers and ionogels, which is different from conventional methods that necessitate specific fillers and polymer matrix. Therefore, from the perspective of improving compatibility, our approach offers an insightful direction for the design of MMMs.

Next, we will answer the comment regarding the comparison of the results with existing literatures. In general, GPU is an indicator of separation efficiency, which is very important in industrial applications. NIGMs with tunable 3D-interconnected ultrafast transport channels possess a boosted CO₂ permeance of 211.0 GPU and an ultrahigh CO₂/N₂ and CO₂/CH₄ selectivity of up to 151 and 47. Therefore, from the perspective of separation efficiency, the GPU data of our NIGMs is meaningful and may attract industrial applications.

We agree with the reviewer that Barrer is more appropriate for comparing the performance of membrane materials, as it eliminates variations due to differences in membrane thickness. Following the reviewer's suggestions, we have revised the comparison plots to elucidate the advantages of NIGMs from two aspects. First, we have compared the CO₂ permeability versus CO₂/N₂ selectivity improvement for NIGMs and reported MMMs, circumventing the confounding effects of low-permeability polymeric matrices on comparison results. The NIGMs exhibit a remarkable 1558% enhancement in CO₂ permeability and a notable 287% increase in CO₂/N₂ selectivity in comparison to GMs (Fig. 3e). In contrast, few MMMs exhibit a concurrent boost in selectivity and permeability, in regardless of the types of nanomaterials employed, including MOFs, COFs, POPs, zeolites, nanosheets and nanofibers (Table S1). Therefore, our nanofiber-interwoven skeleton within polymeric gel matrix holds great potential in improving permeance and selectivity for CO₂ separation simultaneously. Second, we have added the performance comparison plots in the revised manuscript to emphasize permeability and selectivity. The CO₂ permeability of our NIGMs is 844.0 Barrer and our NIGMs surpass the widely accepted the 2019 upper bound limit for CO₂/N₂ separation, which stands in contrast to that most reported GMs and MMMs are below 2019 upper bound limit (Supplementary Fig. 21a). Moreover, our NIGMs also manifest great competitiveness in CO₂ permeability and CO₂/CH₄ selectivity that surpass the 2008 upper bound limit, positioning them among the state-of-the-art MMMs (Supplementary Fig. 21b).

Finally, we have added some recently reported relevant membranes in the figures and supplementary tables. Briefly, supplementary tables 2 and 3 have updated 18 kinds of pure gel membranes, 13 kinds of mixed-matrix gel membranes and 14 kinds of mixed-matrix membranes.

We believe these revisions would help to improve the quality of the manuscript.

Fig. 3e Comparison of the NIGMs and reported MMMs in terms of the improvement of CO₂ permeability and CO₂/N₂ selectivity.

Supplementary Fig. 21 (a) Comparative evaluation of NIGMs against reported GMs and reported MMMs in terms of CO₂ permeability and CO₂/N₂ selectivity. (b) Comparative evaluation of NIGMs against reported GMs and reported MMMs of CO₂ permeability and CO₂/CH₄ selectivity.

Comment 2: The authors refer to a 44-fold increase in permeance (page 5), which seems highly specific. Compared to what? Instead of broadly referring to “conventional polymeric membranes,” they should provide specific examples and references to clarify the comparison.

Response and revision: The comparison object in this sentence is the earliest commercial polymeric membranes, including polyimide (*J. Membr. Sci.* **2003**, 211, 311–334; *J. Membr. Sci.* **2008**, 313, 170–181), cellulose acetate (*J. Membr. Sci.* **1989**, 47, 301–332), polysulfone (*Macromolecules* **1992**, 25, 3424–3434) and polycarbonates (*J. Polym. Sci., Part B: Polym. Phys.* **1987**, 25, 1999–2026). The CO₂ permeability of these membranes is only 4.2 to 10 Barrer (*Polymer* **2013**, 54, 4729–4761).

Following the reviewer’s suggestion, we have added the types of these polymer membranes as well as references in the revised manuscript.

Fig. 1 Design concept and merits of nanofiber-interwoven gel membranes (NIGMs). (a) Bio-inspired design of NIGMs for efficient CO₂ separation. Learning from rapid propagation of signals along the 3D densely-stacked cardiacmuscle fibers, NIGMs are imparted with CNT-interwoven interconnected transport channels and highly-selective CO₂-philic gel matrix. (b) Schematic illustration of structure and performance evolution of NIGMs with conventional polymeric membranes, including polyimide^{22,23}, cellulose acetate^{24,25}, polysulfone^{25,26} and polycarbonates^{25,27}. As comparisons, NIGMs show a prominent boost in permeance and selectivity simultaneously, both of which are much better than that of conventional counterparts.

In **Page 5**: “Consequently, our NIGMs are imparted with preferential selectivity and permeance in CO₂ separation at the same time, and particularly feature even dozens of times higher CO₂ permeance than that of traditional polymeric membranes, including polyimide^{22,23}, cellulose acetate^{24,25}, polysulfone^{25,26} and polycarbonates^{25,27} (Fig. 1b).”

References

- Vu, D. Q., Koros, W. J., Miller, S. J. Mixed matrix membranes using carbon molecular sieves: I. Preparation and experimental results. *J. Membr. Sci.* **211**, 311–334 (2003).
- Zhang, Y. et al. Gas permeability properties of Matrimid® membranes containing the metal-organic framework Cu–BPY–HFS. *J. Membr. Sci.* **313**, 170-181 (2008).

24. Puleo, A. C., Paul, D. R., Kelley, S. S. The effect of degree of acetylation on gas sorption and transport behavior in cellulose acetate. *J. Membr. Sci.* **47**, 301–332 (1989).
25. Sanders, D. F. et al. Energy-efficient polymeric gas separation membranes for a sustainable future: A review. *Polymer* **54**, 4729-4761 (2013).
26. Aitken, C. L., Koros, W. J., Paul, D. R. Effect of structural symmetry on gas transport properties of polysulfones. *Macromolecules* **25**, 3424–3434 (1992).
27. Muruganandam, N., Koros, W. J., Paul, D. R. Gas sorption and transport in substituted polycarbonates. *J. Polym. Sci., Part B: Polym. Phys.* **25**, 1999-2026 (1987).

Comment 3: Since permeance depends on membrane thickness, it is crucial to state the membrane thickness used and what it is compared to when reporting GPU values. A CO₂ permeance of 211 GPU is relatively high, but whether it is truly exceptional depends on the benchmark used. A comparison with lower-permeability polymers may make the results seem extraordinary, but a more balanced evaluation is needed.

Response and revision: Thanks so much for your constructive suggestion. In response to the reviewer’s suggestions, we have included the membrane thickness in the figure legend and have revised the comparison plots to elucidate the advantages of NIGMs from two aspects by eliminating the impact of membrane thickness.

First, we have compared the CO₂ permeability versus CO₂/N₂ selectivity improvement for NIGMs and reported MMMs containing high-permeable polymers and nanomaterials, circumventing the confounding effects of low-permeability polymeric matrices on comparison results. The NIGMs exhibit a remarkable 1558% enhancement in CO₂ permeability and a notable 287% increase in CO₂/N₂ selectivity in comparison to GMs (Fig. 3e). In contrast, few MMMs exhibit a concurrent boost in selectivity and permeability, in regardless of the types of nanomaterials employed, including MOFs, COFs, POPs, zeolites, nanosheets and nanofibers (Table S1). Therefore, our nanofiber-interwoven skeleton within polymeric gel matrix holds great potential in improving permeance and selectivity for CO₂ separation simultaneously.

Second, we have added the performance comparison plots in the revised manuscript to emphasize permeability and selectivity of NIGMs. The CO₂ permeability of our NIGMs is 844.0 Barrer and our NIGMs also surpass the widely accepted the 2019 upper bound limit for CO₂/N₂ separation, which stands in contrast to that most reported GMs and MMMs are below 2019 upper bound limit (Supplementary Fig. 21a). Moreover, our NIGMs also manifest great competitiveness in CO₂ permeability and CO₂/CH₄ selectivity that surpass the 2008 upper bound limit, positioning them among the state-of-the-art MMMs (Supplementary Fig. 21b). Thus, both GPU and Barrer values demonstrate the impressive CO₂ separation performance of our NIGMs.

The manuscript has been revised accordingly.

Fig. 3e Comparison of the NIGMs and reported MMMs in terms of the improvement of CO₂ permeability and CO₂/N₂ selectivity.

Supplementary Fig. 21 (a) Comparative evaluation of NIGMs against reported GMs and reported MMMs in terms of CO₂ permeability and CO₂/N₂ selectivity. (b) Comparative evaluation of NIGMs against reported GMs and reported MMMs of CO₂ permeability and CO₂/CH₄ selectivity.

Comment 4: I strongly advise the authors to remove over-emphatic statements such as “extraordinary CO₂ separation.” Performance evaluation depends on multiple parameters, and while the results are very good, calling them extraordinary is an overstatement. Even in Table S1 and Figure 5, some reported results are, at best, comparable to existing data.

Response and revision: Many thanks for the reviewer’s reminder. Following the reviewer’s suggestions, we have carefully checked and modified the over-emphatic statements, and substantially revised the manuscript. Specifically, the description “unprecedented CO₂ separation” has been revised to “efficient CO₂ separation” (page 1, line 2 and line 13). “Extraordinary CO₂ separation performance” has been modified to “excellent CO₂ separation performance” (page 9, line 4; page 9, line 20; page 17, line 3). Additionally, “exceptional CO₂ permeance” has been updated to “boosted CO₂ permeance” (page 3, line 20; page 9, line 8).

Comment 5: On page 8, the authors refer to “exceptional permeability,” stating that it is “16.6 times higher than conventional GMs with a permeance of 12.7 GPU.” Can they provide references and evidence to confirm that no other materials achieve similar results? This is another very bold statement.

Response and revision: Thank the reviewer for this insightful comment. In the manuscript, “conventional GMs” refers to our lab-prepared PEG gel membranes without CNT-interwoven skeleton. Following the reviewer’s suggestion, we have comprehensively revised over-emphatic statements throughout the manuscript, including excision of the word “conventional”, as well as substitution of “exceptional CO₂ permeance” with “boosted CO₂ permeance” in the revised manuscript.

Additionally, we have added the CO₂ separation performance of the gel membranes reported in recent 5 years as comparison in Table R1. The majority of reported GMs exhibit an unsatisfactory CO₂ permeance in the range of 1.3~10.3 GPU, accompanied with a CO₂/N₂ selectivity between 15 and 63. Although a minority of emerging GMs with ultra-thin gel layer exhibit better CO₂ permeance (778~6100 GPU) than our NIGMs, they often suffer from unsatisfactory selectivity (15~22), far away from our NIGMs (*J. Membr. Sci.*, **2022**, 663, 121032; *Ind. Eng. Chem. Res.*, **2014**, 53, 20064).

Table R1. Comparison between our work and the gel membranes reported in recent 5 years.

	Year	Materials type	CO ₂ permeance (GPU)	CO ₂ permeability (Barrer)	CO ₂ /N ₂ selectivity	CO ₂ /CH ₄ selectivity
ACS Sustainable Chem. Eng., 2020, 8, 5954	2020	pure GMs	0.2	20.4	86.8	118.6
Sep. Purif. Technol., 2020, 250, 117201	2020	pure GMs	2.9	292	25.4	14.5
Sep. Purif. Technol., 2021, 270, 118812	2021	pure GMs	1.3	66.9	26.7	11.4
J. Mater. Chem. A, 2022,10, 4695-4702	2022	pure GMs	3.5	319.1	35.9	16.1
Ind. Eng. Chem. Res., 2022, 61, 4648-4658	2022	pure GMs	3.5	1421	27	\
J. Membr. Sci., 2022, 663, 121032	2022	pure GMs	778	466.8	15	\
J. Membr. Sci., 2022, 660, 120837	2022	pure GMs	6.5	456.4	61.4	18.1
J. Membr. Sci., 2023, 685, 121912	2023	pure GMs	5.6	2240	43	\
Angew. Chem. Int. Ed., 2024, 63, e202411270	2024	pure GMs	2.0	186.4	61.7	24.6
J. Membr. Sci., 2024, 695, 122482	2024	pure GMs	10.3	3100	43	\
J. Membr. Sci., 2024, 711, 123200	2024	pure GMs	2~10	500~2920	61	\
Sep. Purif. Technol., 2024, 331, 125591	2024	pure GMs	4.3	866	31	\
Sep. Purif. Technol., 2024, 331, 125591	2024	pure GMs	2.3	464	63	\
Sep. Purif. Technol., 2025, 359, 130499	2025	pure GMs	98.2	913.3	31	\
Sep. Purif. Technol., 2025, 362, 131916	2025	pure GMs	3.1	310	62	\
Ind. Eng. Chem. Res., 2021, 60, 12640	2021	Mixed-Matrix GMs	3.1	920	20	\

Ind. Eng. Chem. Res., 2021, 60, 12698	2021	Mixed-Matrix GMs	119	535	23	\
Membranes, 2021, 11, 998	2021	Mixed-Matrix GMs	0.8	98	12	7
Polym. J., 2021, 53, 137	2021	Mixed-Matrix GMs	4.5	1362	22	\
J. Membr. Sci., 2022, 660, 120841	2022	Mixed-Matrix GMs	0.7	90.6	27.8	10.5
J. Membr. Sci., 2023, 683, 121818	2023	Mixed-Matrix GMs	10.2	408	97.2	\
J. Membr. Sci., 2023, 683, 121818	2023	Mixed-Matrix GMs	8.5	382.5	27.3	\
J. Membr. Sci., 2023, 683, 121818	2023	Mixed-Matrix GMs	5.9	266.3	45.1	\
J. Membr. Sci., 2023, 685, 121938	2023	Mixed-Matrix GMs	4.8	479	30	17
Angew. Chem. Int. Ed., 2024, 63, e202315607	2023	Mixed-Matrix GMs	28.1	5608	39	\
ACS Appl. Polym. Mater., 2025, 7, 5944	2025	Mixed-Matrix GMs	7.2	725	40	22
Our work (GMs)	2025	pure GMs	12.7	52.1	74	22
Our work (NIGMs)	2025		211.0	844.0	151	47

Comment 6: I agree with the statement that the reported results are very good, but a fairer comparison with other polymers and MMMs could be achieved by including permeability data in Robeson plots and reporting values in Barrer. While GPU is useful for industrial applications, Barrer is more appropriate for comparing materials, as it eliminates variations due to differences in membrane thickness.

Response and revision: This is an insightful comment. In general, GPU is an indicator of separation efficiency, which is very important in industrial applications. NIGMs with tunable 3D-interconnected ultrafast transport channels possess a boosted CO₂ permeance of 211.0 GPU and an ultrahigh CO₂/N₂ and CO₂/CH₄ selectivity of up to 151 and 47. Therefore, from the perspective of separation efficiency, the GPU data of our NIGMs is meaningful and may attract industrial applications.

As observed by the reviewer, we lacked a comparative evaluation of NIGMs against reported GMs and reported MMMs in terms of CO₂ permeability. To address this deficiency, we have added the performance comparison plots in the revised manuscript to emphasize permeability and selectivity. The CO₂ permeability of our NIGMs is 844.0 Barrer and our NIGMs surpass the widely accepted the 2019 upper bound limit for CO₂/N₂ separation, which stands in contrast to that most reported GMs and MMMs are below 2019 upper bound limit (Supplementary Fig. 21a). Moreover, our NIGMs also manifest great competitiveness in CO₂ permeability and CO₂/CH₄ selectivity that surpass the 2008 upper bound limit, positioning them among the state-of-the-art MMMs (Supplementary Fig. 21b).

Moreover, we have added some recently reported relevant membranes in these figures and supplementary tables. Briefly, supplementary tables 2 and 3 have updated 18 kinds of pure gel membranes, 13 kinds of mixed-matrix gel membranes and 14 kinds of mixed-matrix membranes. The manuscript has been revised accordingly.

Supplementary Fig. 21 (a) Comparative evaluation of NIGMs against reported GMs and reported MMMs in terms of CO₂ permeability and CO₂/N₂ selectivity. (b) Comparative evaluation of NIGMs against reported GMs and reported MMMs of CO₂ permeability and CO₂/CH₄ selectivity.

In Page 16: “To further underscore the inherent permeability advantages of our NIGMs, we conducted a rigorous comparative analysis of CO₂ permeability against state-of-the-art membranes. The CO₂ permeability of our NIGMs is 844.0 Barrer, with CO₂/N₂ selectivity surpassing the 2019 Robeson upper bound and CO₂/CH₄ selectivity exceeding the 2008 Robeson upper bound (Supplementary Fig. 21).”

Comment 7: If the permeation mechanism is mainly based on gas diffusion and molecular sieving, I would expect the materials to exhibit high porosity. However, a key aspect missing from this study is porosity assessment using low-temperature isothermal adsorption (e.g., N₂ @ 77 K and/or CO₂ @ 273 K). This is particularly relevant since the authors claim the formation of a dense, defect-free layer, which may not be as porous as suggested.

Response and revision: In response to the reviewer’s suggestions, we have measured the nitrogen adsorption-desorption isotherms of pure GMs and NIGMs at 77 K. As shown in Supplementary Fig. 3, both pure GMs and NIGMs exhibit typical Type II isotherms, suggesting unrestricted monolayer-multilayer adsorption occurring on non-porous or macroporous materials. Furthermore, the BET surface areas of pure GMs and NIGMs are determined to be 0.5998 m²·g⁻¹ and 1.6251 m²·g⁻¹, respectively, which are two orders of magnitude lower than those of reported nanoporous materials (*Nat. Chem. Eng.*, **2024**, 1, 411; *Nat. Commun.*, **2022**, 13, 1427). These results confirm that the NIGMs possess a dense and defect-free structure without apparent voids or defects, rather than a nanoporous structure.

In our work, the CO₂ transport and separation mechanism within NIGMs follows a classic solution-diffusion model rather than molecular sieving effects. The excellent separation performance of NIGMs mainly originates from the function fusion of CNT-interwoven skeleton and defect-free CO₂-philic gel. On the one hand, CNT-interwoven skeleton can not only serve as ultrafast continuous transport pathways to boost CO₂ permeance, but also disrupt the stacking of polymer chains within gel networks for leading to heightened mobility of polymer chains and increased free volume for facilitating CO₂ diffusion and permeance. On the other hand, the in-situ formed CO₂-philic gel matrix has decent interfacial compatibility with CNT skeleton for avoiding non-selective defects, meanwhile harnessing its abundant polar ethylene oxide units to facilitate CO₂ solubility over N₂ for attaining enhanced selectivity.

We have included and discussed these results in the revised manuscript.

Supplementary Fig. 3 Nitrogen adsorption-desorption isotherms of pure GMs and NIGMs at 77 K.

In **Page 7**: “The nitrogen adsorption-desorption isotherms results reveal that NIGMs exhibit typical Type II isotherms with a BET surface area value of merely 1.6251 m²·g⁻¹, confirming that the NIGMs possess a dense and defect-free structure (Supplementary Fig. 3).”

Comment 8: The last paragraph should be titled “Conclusions” rather than “Discussion.”

Response and revision: Thanks for the reviewer’s reminder. Following the reviewer’s suggestion, the title of the last paragraph has been revised from “Discussion” to “Conclusions”.

Response to the comments by Reviewer 3

Comment 1: This manuscript reports an innovative and highly impactful strategy for fabricating mixed matrix membranes (MMMs) with unprecedented CO₂ separation performance. The authors propose a bio-inspired "nanofiber-interwoven gel membrane" (NIGM) architecture that integrates a carbon nanotube (CNT) network and a CO₂-philic polymer gel matrix via a photothermal-triggered in-situ gelation process. The result is a defect-free membrane that achieves both exceptional CO₂ permeance (211 GPU) and CO₂/N₂ selectivity (151), outperforming the Robeson 2019 upper bound and most reported MMMs. The study is well-structured, systematically executed, and thoroughly benchmarked. It introduces a fabrication method of general utility and presents significant advancement in gas separation membrane design. I believe this work is suitable for publication in Nature Communications after addressing a few minor comments.

Response: We are pleased that the reviewer found our work innovative and recommended for revision. The insightful comments raised by the reviewer help improve the quality of our paper.

Comment 2: Phrases such as “cardiacmuscle-inspired ultrafast transport channels” may appear overly metaphorical for a materials science audience. I suggest rephrasing these in more conventional membrane science terms to improve scientific clarity.

Response and revision: We appreciate the reviewer bringing this matter to our attention. Following the reviewer’s suggestions, we have carefully checked and modified the metaphorical and unscientific phrases, and substantially revised the manuscript. Specifically, “cardiacmuscle-inspired ultrafast transport channels” has been renamed as “tunable 3D-interconnected transport channels” (page 1, line 1 and line 12).

Comment 3: Please briefly discuss the potential for scaling up the photothermal-triggered fabrication method. How feasible is sunlight-based gelation for industrial-scale membrane production? Could alternative irradiation sources be employed?

Response and revision: Following the reviewer’s suggestions, we have successfully achieved the fabrication of 30-cm-diameter NIGMs with uniform thickness (Supplementary Figs. 4 and 5). Initially, a 30 cm-diameter CNT-interwoven photothermal reactor was fabricated using a customized large vacuum filtration device (Supplementary Fig. 5a). Subsequently, the CNT-interwoven reactor was uniformly sprayed with gel precursors ($0.13 \text{ g}\cdot\text{cm}^{-2}$) by spray gun. The precursor-infused photothermal reactor was then exposed to a multi-Xenon lamp system (Supplementary Fig. 5b), enabling large-area fabrication of NIGMs through photothermal-triggered in-situ gelation method. Moreover, we also proposed a viable continuous route to further scale up the photothermal-triggered fabrication method by coupling vacuum filtration with spraying, as illustrated in Supplementary Fig. 6. To make sunlight-based gelation feasible for industrial-scale membrane production, we employed a multi-Xenon lamp array system to expand the projected solar irradiation area, thus paving the way for industrial-scale NIGMs production via our photothermal-triggered gelation method.

In addition to the sunlight-based gelation, we also investigated the photothermal effects of the CNT-interwoven confined reactor under irradiation of multi-UV light system (Supplementary Fig. 7). Given that the CNT-interwoven reactor exhibits a 98.6% absorption across the ultraviolet spectrum (Supplementary Fig. 1), its surface temperature rapidly escalates to $78.6 \text{ }^\circ\text{C}$ within 120 seconds under UV exposure ($1500 \text{ W}\cdot\text{m}^{-2}$). This rapid photothermal response of CNT-interwoven reactor provides a thermal basis for initiating in-situ gelation of precursors, demonstrating the feasibility of employing multi-UV light systems for scalable NIGMs production via our photothermal-triggered in-situ gelation method.

We have included and discussed these results in the revised manuscript.

Supplementary Fig. 4 30-cm-diameter NIGMs with uniform thickness.

Supplementary Fig. 5 Digital images of (a) customized large vacuum filtration device and (b) multi-Xenon lamp system.

Supplementary Fig. 6 Schematic diagram of the industrial-scale fabrication of NIGMs through photothermal-triggered in-situ gelation method.

Supplementary Fig. 7 Dynamic curve of temperature of the CNT-interwoven photothermal confined reactor under irradiation of multi-UV light system.

In **Page 7**: “Moreover, this photothermal-triggered in-situ gelation method can be easily scaled up for constructing 30 cm-diameter NIGMs with uniform thickness of about 4 μm (Supplementary Figs. 4-7), implying the potential for large-scale fabrication of NIGMs.”

In **Supplementary materials**: “The 30-cm-diameter NIGMs were fabricated as follows: Initially, a 30 cm-diameter CNT-interwoven photothermal reactor was fabricated using a customized large vacuum filtration device (Supplementary Fig. 5a). Subsequently, the CNT-interwoven reactor was uniformly sprayed with gel precursors ($0.13 \text{ g}\cdot\text{cm}^{-2}$) by spray gun. The precursor-infused photothermal reactor was then exposed to a multi-Xenon lamp system (Supplementary Fig. 5b), enabling large-area fabrication of NIGMs through photothermal-triggered in-situ gelation method. Moreover, we also proposed a viable route to further scale up the photothermal-triggered fabrication method by coupling vacuum filtration with spraying, as illustrated in Supplementary Fig. 6. To make sunlight-based gelation feasible for industrial-scale membrane production, we employed a multi-Xenon lamp array system to expand the projected solar irradiation area, thus paving the way for industrial-scale NIGMs production via our photothermal-triggered gelation method.

In addition to the sunlight-based gelation, we also investigated the photothermal effects of the CNT-interwoven confined reactor under irradiation of multi-UV light system (Supplementary Fig. 7). Given that the CNT-interwoven reactor exhibits a 98.6% absorption across the ultraviolet spectrum (Supplementary Fig. 1), its surface temperature rapidly escalates to $78.6 \text{ }^\circ\text{C}$ within 120 seconds under UV exposure ($1500 \text{ W}\cdot\text{m}^{-2}$). This rapid photothermal response of CNT-interwoven reactor provides a thermal basis for initiating in-situ gelation of precursors, demonstrating the feasibility of employing multi-UV light systems for scalable NIGMs production via our photothermal-triggered in-situ gelation method.”

Comment 4: While the diffusion-dominated mechanism is well supported by data, inclusion of additional discussion on gas pathway modeling or free volume distributions (even qualitatively) would enhance mechanistic understanding.

Response and revision: We agree with the reviewer that CO_2 transport and separation mechanism within NIGMs is heavily dictated by the diffusion process. Following the reviewer’s suggestions, we have calculated fractional free volume for the NIGMs and pure GMs via molecular dynamics simulation. The results, as portrayed in Supplementary Fig. 17, unveil a fractional free volume of 6.9% for the pure GMs, contrasting with the higher fractional free volume of 13.9% exhibited by the NIGMs. In addition to the fractional free volume, we also calculated diffusion coefficients of CO_2 nearby PEG matrix and nearby CNT within NIGMs, respectively (Supplementary Fig. 18). Significantly, the inclusion of the CNT triggers a substantial elevation in gas diffusivity, escalating from $5.23 \times 10^{-8} \text{ cm}^2\cdot\text{s}^{-1}$ for CO_2 nearby PEG matrix to an impressive $1.16 \times 10^{-5} \text{ cm}^2\cdot\text{s}^{-1}$ for CO_2 nearby CNT within NIGMs. Above simulation results harmonize seamlessly with outcomes derived from measured diffusivity. The CNT-interwoven skeleton of NIGMs disrupts the stacking of polymer chains within gel networks and engenders augmented mobility of polymer chains, resulting in a substantial augmentation in both the fractional free volume and diffusion coefficients, which, in turn, facilitates the diffusion of CO_2 within the densely-packed CNT-interwoven skeleton.

We have included and discussed these results in the revised manuscript.

Supplementary Fig. 17 Fractional free volume for the NIGMs and pure GMs via molecular dynamics simulation.

Supplementary Fig. 18 (a) Mean square displacement and (b) diffusion coefficients of CO₂ nearby PEG matrix and nearby CNT within NIGMs.

In **Page 13**: “To further reveal CO₂ transport mechanism, we calculated fractional free volume for the NIGMs and pure GMs via molecular dynamics simulation. The results show that pure GMs have a fractional free volume of 6.9%, contrasting with the higher fractional free volume of 13.9% for the NIGMs (Supplementary Fig. 17). In addition to the fractional free volume, we also calculated diffusion coefficients of CO₂ nearby PEG matrix and nearby CNT within NIGMs (Supplementary Fig. 18). Significantly, the inclusion of the CNT triggers a substantial elevation in gas diffusivity, escalating from $5.23 \times 10^{-8} \text{ cm}^2 \cdot \text{s}^{-1}$ for CO₂ nearby PEG matrix to an impressive $1.16 \times 10^{-5} \text{ cm}^2 \cdot \text{s}^{-1}$ for CO₂ nearby CNT within NIGMs. Above simulation results harmonize seamlessly with outcomes derived from measured diffusivity.”

Comment 5: Consider adding commentary on how these membranes might perform under varied humidity or temperature cycling, as this would be relevant for real-world CO₂ capture applications.

Response: We highly appreciate the reviewer's professional comments and sharp scientific insights. In response to the reviewer's suggestions, we have investigated the influence of the humidity and temperature on the CO₂ separation performance of our NIGMs. As the relative humidity (RH) value is elevated from 0% to 70%, the CO₂ permeance of the NIGMs increases by 8.5%, attaining $229.3 \pm 1.6 \text{ GPU}$, while its CO₂/N₂ selectivity decreases by 45.6% to only 82.5 ± 2.7 (Supplementary Fig. 23). It is noteworthy that our NIGMs demonstrate dynamic responsiveness to ambient humidity cycling, with permeance and selectivity gradually restoring to initial values upon re-exposure to dry environments (RH = 0%).

In addition to the humidity, we have also measured the change of CO₂ separation performance of NIGMs under different temperatures. The CO₂ permeance of NIGMs increases monotonically with temperature, reaching 297.4 ± 8.3 GPU at 40 °C and further escalating to 406.1 ± 11.6 GPU at 60 °C (Supplementary Fig. 24). This is because the mobility of CO₂ molecules enhances at elevated temperatures, thereby augmenting the driving force for diffusion. Moreover, the increase in temperature renders the PEG chains more flexible, creating additional free volume cavities for molecular transport. However, CO₂/N₂ selectivity of NIGMs displays a decrease trend as temperature increases, from 151.2 ± 3.4 at 25 °C to 73.1 ± 5.9 at 60 °C. This can be attributed to the decreased CO₂ solubility within the PEG matrix at higher temperatures, which decreases both CO₂/N₂ solubility selectivity and overall separation selectivity (*J. Membr. Sci.*, **2007**, 291, 131-139; *J. Membr. Sci.*, **2013**, 437, 286-297; *J. Membr. Sci.*, **2014**, 460, 62-70).

Following the reviewer's suggestions, we have included and discussed these results in the revised manuscript.

Supplementary Fig. 23 (a) CO₂ permeance and (b) selectivity of NIGMs varied humidity cycling.

Supplementary Fig. 24 CO₂ separation performance of NIGMs under different temperatures.

In **Page 17**: “In real-world CO₂ capture applications, some harsh conditions such as high humidity and temperature environments are often encountered. We further investigated the influence of the humidity and temperature on the CO₂ separation performance of our NIGMs. NIGMs display enhanced CO₂ permeance but diminished selectivity in damp environments. As the relative humidity (RH) value is elevated from 0% to 70%, the CO₂ permeance of the NIGMs increases by 8.5%, attaining 229.3 GPU, while its CO₂/N₂ selectivity decreases by 45.6% to only 82.5 (Supplementary Fig. 23). It is noteworthy that our NIGMs demonstrate dynamic responsiveness to ambient humidity cycling, with permeance and selectivity gradually restoring to initial values upon re-exposure to dry environments (RH = 0%). In analogy with the humidity, the CO₂ permeance of NIGMs increases monotonically with temperature, reaching 297.4 GPU at 40 °C and further escalating to 406.1 GPU at 60 °C (Supplementary Fig. 24). However, CO₂/N₂ selectivity of NIGMs displays a decrease trend

as temperature increases, from 151.2 at 25 °C to 73.1 at 60 °C. This can be attributed to the decreased CO₂ solubility within the PEG matrix at higher temperatures, which decreases both CO₂/N₂ solubility selectivity and overall separation selectivity. To recap, our NIGMs still possess good CO₂ separation performance even under high humidity and temperature environments.”